# Hormone-sensitive lipase couples intergenerational sterol metabolism to reproductive success

Christoph Heier[1,2]*, Oskar Knittelfelder[3†], Harald F Hofbauer[1,2],
Wolfgang Mende[1], Ingrid Pörnbacher[1], Laura Schiller[1], Gabriele Schoiswohl[1,4],
Hao Xie[1], Sebastian Grönke[5‡], Andrej Shevchenko[3], Ronald P Kühnlein[1,2,4,5]

[1]Institute of Molecular Biosciences, University of Graz, Graz, Austria; [2]BioTechMed-Graz, Graz, Austria; [3]Max Planck Institute of Molecular Cell Biology and Genetics, Dresden, Germany; [4]Field of Excellence BioHealth - University of Graz, Graz, Austria; [5]Max Planck Institute for Biophysical Chemistry, Göttingen, Germany

**Abstract** Triacylglycerol (TG) and steryl ester (SE) lipid storage is a universal strategy to maintain organismal energy and membrane homeostasis. Cycles of building and mobilizing storage fat are fundamental in (re)distributing lipid substrates between tissues or to progress ontogenetic transitions. In this study, we show that Hormone-sensitive lipase (Hsl) specifically controls SE mobilization to initiate intergenerational sterol transfer in *Drosophila melanogaster*. Tissue-autonomous Hsl functions in the maternal fat body and germline coordinately prevent adult SE overstorage and maximize sterol allocation to embryos. While Hsl-deficiency is largely dispensable for normal development on sterol-rich diets, animals depend on adipocyte Hsl for optimal fecundity when dietary sterol becomes limiting. Notably, accumulation of SE but not of TG is a characteristic of Hsl-deficient cells across phyla including murine white adipocytes. In summary, we identified Hsl as an ancestral regulator of SE degradation, which improves intergenerational sterol transfer and reproductive success in flies.

*For correspondence:
christoph.heier@uni-graz.at

Present address: † Institute of Chemistry, Humboldt University of Berlin, Berlin, Germany; ‡ Max Planck Institute for Biology of Ageing, Cologne, Germany

Competing interests: The authors declare that no competing interests exist.

## Introduction

Lipids fulfill essential functions as energy carriers, membrane components, and signal molecules. The constant supply and recycling of lipids are therefore fundamental for maintaining organismal vitality. However, chronic excess of lipids can also lead to lipotoxicity and numerous associated diseases in humans such as obesity, type 2 diabetes, or cardiovascular diseases (*Brookheart et al., 2009*; *Greenberg et al., 2011*). An evolutionarily successful strategy for coping with fluctuations in the availability of lipids is their conversion into neutral esters and storage for future need. In this form, lipids are deposited in so-called intracellular lipid droplets (LDs) (*Walther and Farese, 2012*). If required, the stored lipids are released by enzymatic hydrolysis of the ester bonds, which makes them available for metabolism or transport processes. Triacylglycerol (TG) and steryl ester (SE) are the most common storage lipids in many cells and organisms (*Bartz et al., 2007*; *Fujimoto and Parton, 2011*). TG constitutes a quantitatively significant energy reserve by providing a major reservoir of fatty acids (FAs) for β-oxidation (*Frayn et al., 2006*). Also, the formation of TG prevents lipotoxicity by sequestering excess FAs (*Listenberger et al., 2003*). Likewise, SE formation buffers excess sterols and generates storage reservoirs for hormonal precursors, particularly in steroidogenic tissues (*Luo et al., 2020*; *Shen et al., 2016*).

Cycles of lipid storage and mobilization are an integral part of the (re)distribution of lipid resources between tissues, ontogenetic stages, and from parents to offspring (*Arrese and Soulages, 2010*; *Babin and Gibbons, 2009*). As the lipid storage capacity of most cell types is very limited,

animals use specialized adipose tissues to build up and store substantial TG-rich LDs in times of nutrient excess (*Azeez et al., 2014*). Upon nutritional deprivation, mobilization of this storage lipid depots initiates transfer of FAs to non-adipose tissues for energy production (*Azeez et al., 2014*; *Lafontan and Langin, 2009*). During reproduction, many animals deposit LD-like structures in the milk or in the egg as a means of lipid transfer to the progeny. Effective transfer and subsequent breakdown of these lipids are essential for offspring vitality and also supports developmental transitions between ontogenetic stages (*Beigneux et al., 2006*; *Grönke et al., 2005*; *Lowe et al., 1998*; *Smith et al., 2000*). By initiating mobilization of stored TG or SE, lipid hydrolases act as gatekeepers in lipid transfer processes associated with nutrient deprivation or development (*Grönke et al., 2005*; *Haemmerle et al., 2006*).

The fruit fly *Drosophila melanogaster* (hereafter referred to as *Drosophila*) shares many principles of metabolism and physiology with higher organisms and is gaining increasing popularity as a model organism for research in lipid metabolism and related metabolic diseases (*Musselman and Kühnlein, 2018*). Like mammals, *Drosophila* uses highly specialized tissues for absorption, transport, and storage of lipids and requires complex communication mechanisms between these tissues to maintain lipid homeostasis (*Heier and Kühnlein, 2018*; *Musselman and Kühnlein, 2018*; *Song et al., 2018*). *Drosophila* requires both, de novo lipogenesis and dietary lipid sources for lipid homeostasis (*Garrido et al., 2015*; *Sieber and Thummel, 2009*). In particular, *Drosophila* is unable to de novo synthesize sterol lipids, which are important membrane constituents and precursors for protein modification and molting hormones (*Carvalho et al., 2010*). For efficient growth and development *Drosophila* thus requires molecular mechanisms that efficiently balance fluctuations in sterol availability.

To cope with fluctuations in nutrient availability, *Drosophila* deposits large amounts of storage lipids in the fat body, which shares many lipid metabolic functions with mammalian adipose tissue and liver, and serves as major nutrient reservoir (*Arrese and Soulages, 2010*). Fat body TG constitutes the quantitatively most important organismal energy reservoir upon starvation (*Aguila et al., 2007*; *Grönke et al., 2007*). Fat body lipids may also support lipid anabolic processes in non-adipose tissues such as the ovary, which incorporates large amounts of SE and TG into vitellogenic oocytes (*Parra-Peralbo and Culi, 2011*; *Sieber and Spradling, 2015*). A lipoprotein-based carrier system connects the fat body with non-adipose tissues via the hemolymph (*Palm et al., 2012*). Importantly, enzymatic hydrolysis of TG and SE and the subsequent incorporation of diacylglycerol (DG) and free sterol into hemolymph lipoproteins is a prerequisite for the transfer of stored lipids between *Drosophila* tissues.

Two complementary enzymatic systems control the hydrolysis of TG in the adult *Drosophila* fat body (*Heier and Kühnlein, 2018*). Brummer (Bmm) lipase is the ortholog of mammalian adipose triglyceride lipase (ATGL) and a major regulator of basal and starvation-induced TG lipolysis (*Grönke et al., 2005*). Bmm activity is complemented by a second lipolytic system that is activated by the Adipokinetic hormone (Akh), a peptide with functional similarity to mammalian glucagon (*Gáliková et al., 2015*; *Grönke et al., 2007*). Akh binding to the G-protein-coupled Akh receptor triggers elevations in adipocyte cAMP and $Ca^{2+}$ levels, which promote lipolysis via incomprehensively understood molecular events. In particular, the identity of the *Drosophila* Akh-dependent TG hydrolase(s) is unclear (*Heier and Kühnlein, 2018*). The enzymes controlling the degradation of SE in *Drosophila* are even more enigmatic. The acid lipase family member Magro has been shown to control SE hydrolysis in the enterocytes of the midgut (*Sieber and Thummel, 2012*). However, the role of Magro and other SE hydrolases in sterol distribution between tissues is largely unknown.

A prime candidate enzyme for both, SE and TG hydrolysis in *Drosophila* is Hormone-sensitive lipase (Hsl). Metazoan Hsl enzyme family members, including human and fly Hsl, are characterized by a N-terminal Hsl_N domain (PF06350) and a C-terminal part composed of a α/β hydrolase fold three domain (PF07859) interspersed with a regulatory module. The α/β hydrolase fold three domain is a phylogenetically ancient catalytic domain also present in the prokaryotic enzymes of the bacterial Hsl (bHsl) family (*Langin and Holm, 1993*; *Lass et al., 2011*). Substrate promiscuity covering a wide range of neutral lipids including TG and SE is a common feature of Hsl-related proteins (*Lass et al., 2011*). However, Hsl enzymes from bacteria to mammals have been primarily associated with the degradation of acylglycerols like TG or DG (*Bi et al., 2012*; *Deb et al., 2006*; *Haemmerle et al., 2002a*; *Liu et al., 2017*). In fact, Hsl is a TG and major DG lipase in mammalian adipocytes where it acts in parallel and downstream of ATGL to promote storage lipid breakdown (*Lass et al., 2011*). In addition, mouse Hsl acts as major SE hydrolase in several tissues including liver, intestine, testis, and

adrenal gland (*Kraemer et al., 2004*; *Obrowsky et al., 2012*; *Osuga et al., 2000*; *Sekiya et al., 2008*). Mammalian Hsl activity is stimulated by catabolic hormones such as glucagon or catecholamines (*Lass et al., 2011*). In an effort to understand how lipid hydrolases control inter-tissue lipid homeostasis, we characterized the molecular and physiological functions of the single Hsl ortholog of *Drosophila*. We show that adult Hsl-deficient *Drosophila* exhibit largely normal TG homeostasis, but accumulate excessive SE due to defective SE hydrolysis in their adipocytes. By initiating SE breakdown, Hsl promotes efficient sterol transfer from mothers to progeny and increases the embryonic sterol content during *Drosophila* development to support reproductive success.

## Results

### *Drosophila Hsl* encodes a multifunctional lipid ester hydrolase

To address the molecular function(s) of *Drosophila* Hsl, we expressed a tagged version of the recombinant fly protein (His$_6$-Hsl) in heterologous cultured cells and measured lipid hydrolase activities in cell extracts. Murine Hsl (His$_6$-*Mm*Hsl) was expressed in parallel to serve as a positive control in these assays. Immunoblotting analysis confirmed the ectopic expression of recombinant His$_6$-Hsl, His$_6$-*Mm*Hsl, and of the His$_6$-β-Galactosidase (His$_6$-β-Gal) negative control (*Figure 1A*). Expression of His$_6$-*Mm*Hsl increased cellular hydrolysis rates toward monoacylglycerol (MG), DG, TG, and SE by 6-, 9-, 6-, and 66-fold, respectively, as compared to His$_6$-β-Gal expressing control cell extracts (*Figure 1B*). Expression of fly His$_6$-Hsl elevated cellular hydrolysis rates toward MG, DG, TG, and SE by 14-, 16-, 10-, and 17-fold, respectively, as compared to His$_6$-β-Gal expressing controls indicating a similar substrate spectrum of murine and fly Hsl (*Figure 1B*). To confirm these results in vivo, we engineered transgenic flies and expressed Hsl via the UAS/Gal4 system using the ubiquitous *Act5c-GAL4* driver. Ectopic expression of Hsl increased DG, TG, and SE hydrolase activities of abdominal extracts by 7-, 2-, and 8-fold, respectively, when compared to control samples. In contrast, abdominal MG hydrolase activities were not significantly altered by Hsl expression (*Figure 1C*). Finally, we assessed the subcellular localization of a fluorescently-tagged version of Hsl (Hsl-EGFP) in fat body cells of adult flies and found that Hsl-EGFP concentrated at the surface of LDs (*Figure 1D*). The intensities of the Hsl-EGFP signal at individual droplets were highly variable suggesting enrichment of the protein at a subset of LDs, although this finding awaits confirmation via detection of endogenous Hsl. Taken together, our data show that Hsl is a LD-associated multifunctional lipid ester hydrolase, which implies a function of the enzyme in *Drosophila* storage lipid breakdown.

### Normal TG and energy homeostasis in *Hsl* mutants

To investigate the role of Hsl in organismal lipid metabolism, we used imprecise P-element excision to generate the *Hsl*[1] deletion mutant, which lacks 2.74 kb genomic *Hsl* sequences, including most of the open-reading frame and the start codon (*Figure 2A*, *Figure 2—figure supplement 1*). This results in the absence of detectable *Hsl* mRNA whereas mRNA concentrations of the neighboring genes *Ate1* and *PCNA* are unaffected (*Figure 2—figure supplement 1*). Homozygous *Hsl*[1] mutant animals are viable and were used to address the function of the enzyme in TG metabolism. Inspection of mature adult *Hsl*[1] mutant fat body tissue revealed no apparent difference in the size or abundance of LDs when compared to age-matched controls (*Figure 2B*). Consistently, whole-body TG and DG levels of these *Hsl*[1] mutants were indistinguishable from age-matched controls (*Figure 2C*). Species with 40–50 acyl carbon atoms and 0–3 double bonds accounted for >80% of the TG pool with comparable species distributions in both genotypes (*Figure 2D*). To exclude that Hsl functions in redundancy with other TG lipolytic pathways, we next assessed the expression of genes involved in TG storage regulation. The mRNA concentrations of *bmm*, *Akh*, *perilipin1* (*plin1*), and *perilipin2* (*plin2*) were similar in *Hsl*[1] mutant and control animals arguing against a compensatory transcriptional dysregulation of these genes in response to *Hsl* deficiency (*Figure 2E*). We then combined the *Hsl*[1] mutant allele with loss-of-function mutations of *Akh* or *bmm* and assessed TG levels in the ad libitum fed state and starvation. Unlike *Hsl*[1], the *Akh*[A] and *bmm*[1] alleles were associated with 1.9-fold increased TG levels under ad libitum fed conditions (*Figure 2F*). As previously reported, starvation-induced consumption of TG was impaired in *bmm*[1] mutants and abolished in *Akh*[A] *bmm*[1] double mutants, illustrating the strong genetic interaction between both lipolytic systems (*Grönke et al., 2007*). In contrast, *Hsl*[1] *bmm*[1] or *Hsl*[1] *Akh*[A] double mutants were capable of

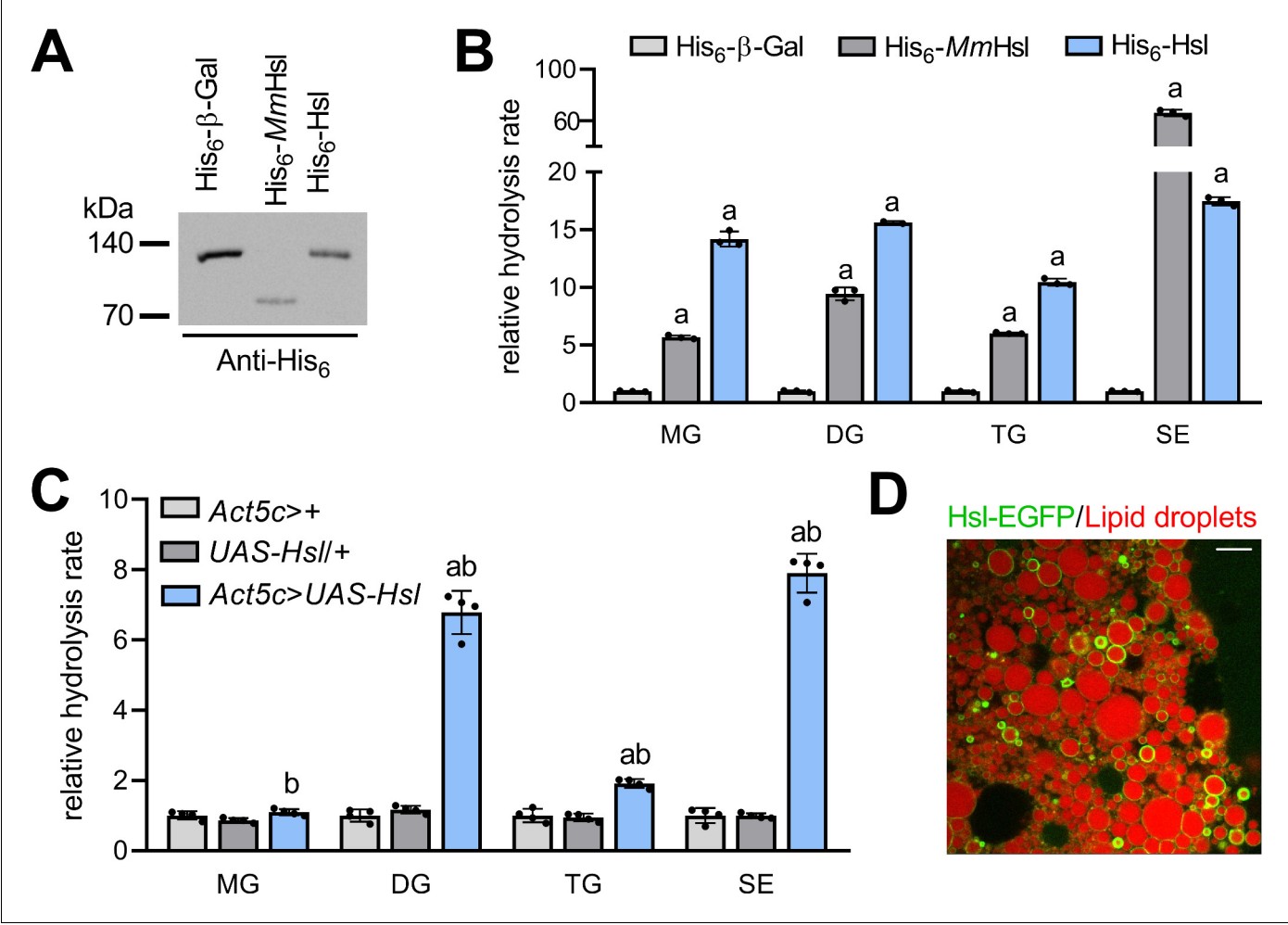

**Figure 1.** Enzyme activities and subcellular localization of *Drosophila* Hsl. (A) Immunoblotting analysis of His₆-tagged proteins in COS-7 cell extracts expressing His₆-Hsl, His₆-*Mm*Hsl, or His₆-β-Gal. (B) Lipid hydrolase activities of recombinant Hsl. Cell extracts expressing His₆-Hsl or His₆-*Mm*Hsl were incubated with different lipids, the release of FAs was measured and normalized to the His₆-β-Gal control (n = 3). (C) Lipid hydrolase activities of *Hsl* gain-of-function flies. *UAS-Hsl* transgene expression was ubiquitously driven in vivo using the *Act5c-GAL4* driver. Lipid hydrolase activities of abdominal extracts were determined as in (B). Flies harboring either the *Act5c-GAL4* transgene or the non-induced *UAS-Hsl* transgene only served as controls and data were normalized to *Act5c-GAL4>+* samples (n = 4). (D) Lipid droplet-associated localization of Hsl-EGFP in fat body cells. Fat body tissue expressing a *UAS-Hsl-EGFP* transgene driven by *Act5c-GAL4* was stained with LipidTOX Deep Red to detect lipid droplets and imaged by confocal fluorescence microscopy. Scale bar: 10 µm. All data are presented as means and SD. Statistical significance was determined by (B) one-way ANOVA (a, p>0.05 vs His₆-β-Gal) and (C) one-way ANOVA (a, p<0.05 *Act5c>UAS-Hsl* vs. *UAS-Hsl/+* and b, p<0.05 *Act5c>UAS-Hsl* vs *Act5c>+*).

starvation-induced TG mobilization and largely resembled *bmm¹* or *Akhᴬ* single mutants in their TG storage phenotypes (***Figure 2F***). Next, we more broadly assessed energy metabolism in *Hsl¹* mutant animals and found levels of free glycerol, free FAs, and glucose unchanged in ad libitum fed *Hsl¹* mutants as compared to age-matched control animals whereas glycogen levels were 26% higher in *Hsl¹* mutants as compared to controls (***Figure 2G***). Microcalorimetry revealed similar heat dissipation rates of *Hsl¹* and control animals under ad libitum fed conditions. Upon fasting, control and *Hsl¹* mutant animals reduced heat dissipation rates by 40% and 46%, respectively, indicating similar whole-body bioenergetics in both genotypes (***Figure 2H***). Also, the response of *Hsl¹* mutants to starvation stress, a sensitive readout of perturbed energy metabolism, was indistinguishable from controls (***Figure 2I***). Taken together, these data argue against a function of *Hsl* in the regulation of TG and energy metabolism in adult *Drosophila* in parallel or downstream of other lipolytic systems.

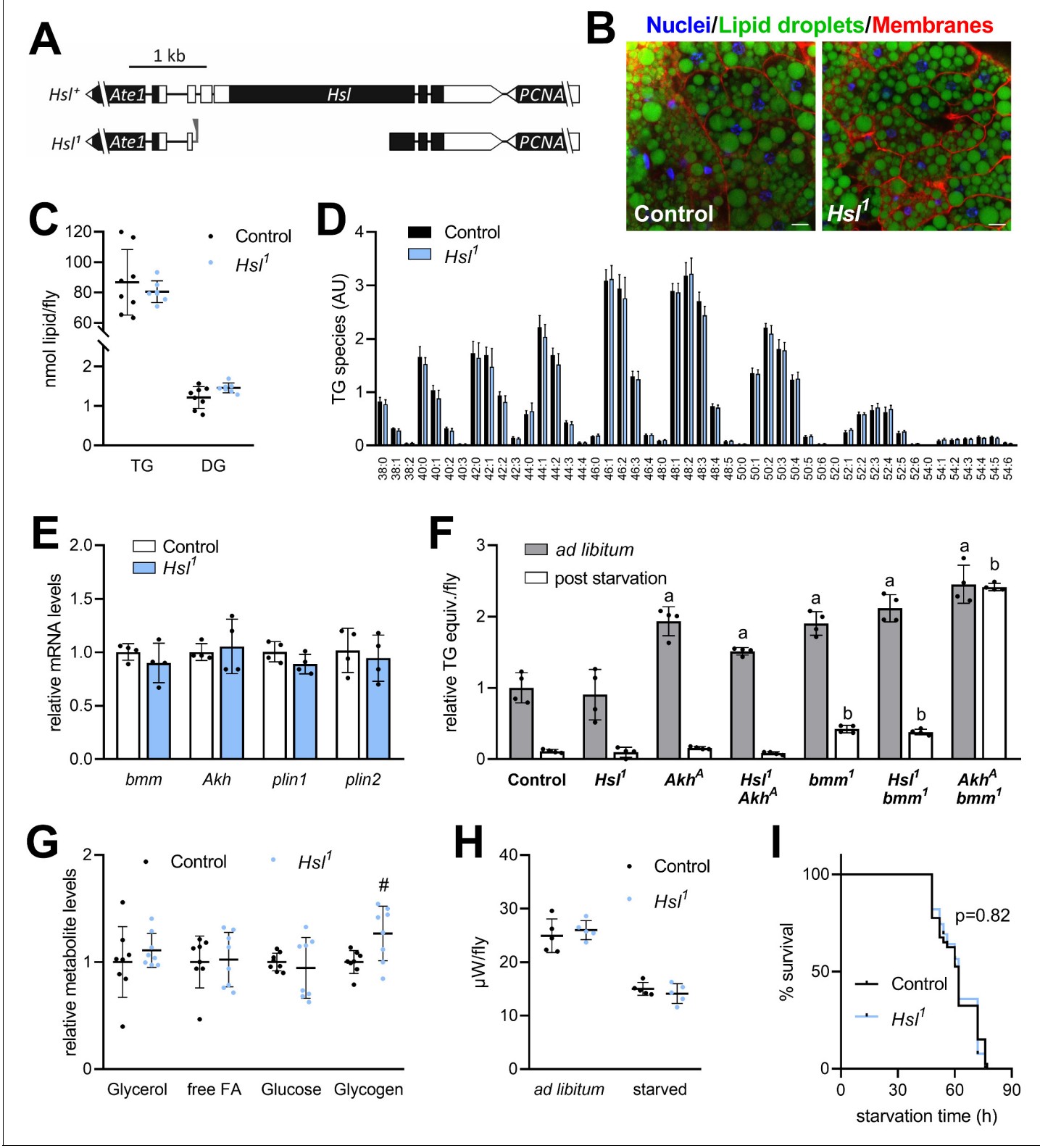

**Figure 2.** Normal TG and energy metabolism in *Hsl1* mutant flies. (A) Organization of the *Hsl* genomic region (including the neighboring genes *Ate1* and *PCNA*), the *Hsl* gene locus and the *Hsl1* deletion mutant allele. Black and white boxes represent coding and non-coding exons, respectively. Residual P-element sequences are indicated by a grey triangle. (B) Abdominal fat body tissue of ad libitum fed control and *Hsl1* mutant animals was stained with Hoechst 33342, Cellmask Deep Red and BODIPY 493/503 to visualize nuclei, cell membranes and LDs, respectively, and imaged by

*Figure 2 continued on next page*

Figure 2 continued

confocal fluorescence microscopy. Scale bars: 10 µm. (C) Whole-body TG and DG levels of ad libitum fed *Hsl[1]* mutant and control animals as determined by shotgun MS (n = 7–8). (D) TG composition of ad libitum fed *Hsl[1]* and control animals as determined by UPLC-MS (n = 4). (E) Lipolytic gene expression in *Hsl[1]* mutants. Relative mRNA concentrations were determined by qPCR and normalized to controls (n = 4). (F) TG equivalents of ad libitum fed and starved *Drosophila* mutants. Flies were fed ad libitum for 7 days and TG equivalents were determined by a coupled colorimetric assay either before or after starvation to death (n = 4). (G) Metabolite levels in ad libitum fed *Hsl[1]* mutant and control flies were measured by colorimetric assays (n = 8). (H) Heat dissipation of ad libitum fed or starved *Hsl[1]* mutant and control flies was determined by microcalorimetry (n = 5). (I) Starvation sensitivity of *Hsl[1]* mutant flies. 7-days-old flies were subjected to starvation and survival was monitored every 2–12 hr (n = 39–40). Data are presented as mean and SD (C, D, E, F, G, H) or Kaplan-Meier curve (I). Statistical significance was determined by (C, D, E, G, H) unpaired *t*-tests (#, p>0.05 compared to control), (F) one-way ANOVA (a, p<0.05 compared to ad libitum fed control; b, p<0.05 compared to starved control) and (I) log-rank test. The online version of this article includes the following figure supplement(s) for figure 2:

**Figure supplement 1.** Molecular characterization of the *Hsl[1]* mutant allele.

## Hsl regulates SE catabolism

The finding that Hsl possesses strong lipolytic activity toward SE in vitro prompted us to investigate sterol metabolism in *Hsl[1]* mutant animals. Compared to controls, SE levels of *Hsl[1]* mutants were similar at 1 day after eclosion, but 2.1- and 2.5-fold higher at 7 days and 14 days, respectively (*Figure 3A*). SE overstorage was also observed in 7-days-old females and males homozygous for the genetically independent *Hsl[b24]* allele (*Figure 3B*; *Bi et al., 2012*). Moreover, neutral SE hydrolase activity in *Hsl[1]* abdominal extracts was 24% lower compared to controls (*Figure 3C*). To address SE turnover more directly, we labeled sterol lipids during the larval period by feeding either [3]H-cholesterol or [14]C-FAs and followed the turnover of labeled lipids after eclosion. [3]H-cholesterol is incorporated into free and esterified sterols and thus allows the simultaneous measurement of both sterol pools. Since the SE pool has a mixed sterol composition (see below), we used [14]C-FA as a complementary tracer to label also ergosteryl and phytosteryl esters. Consistent with reduced SE hydrolysis *Hsl[1]* mutants exhibited slower turnover rates of SE prelabeled with [14]C-FAs or [3]H-cholesterol (*Figure 3D,E*). In contrast, turnover of free [3]H-cholesterol was unaffected by impaired SE catabolism in *Hsl[1]* mutant animals (*Figure 3F*). In line with this observation, 7-days-old *Hsl[1]* mutant animals exhibited unaltered levels of total free sterols when compared to age-matched controls suggesting that defective SE catabolism does not limit free sterols due to dietary sterol supply under ad libitum fed conditions (*Figure 3G*). *Drosophila* is able to use a variety of sterols including cholesterol, phytosterols, and ergosterol. The molecular composition of its sterol pool depends on dietary availability and sterol turnover (*Carvalho et al., 2012*; *Knittelfelder et al., 2020*). We therefore asked whether defective SE catabolism in *Hsl[1]* mutants affects sterol composition. However, both *Hsl[1]* and control animals showed a similar sterol composition dominated by ergosterol and β-sitosterol with lower levels of campesterol and stigmasterol and traces of brassicasterol, desmo-/zymosterol, and lanosterol (*Figure 3G*). To test if SE accumulation in *Hsl[1]* mutants is accompanied by misexpression of other sterol metabolic genes, we assessed mRNA concentrations of the sterol-responsive transcription factor *Hr96*, the Niemann-Pick-related genes *Npc1a*, *Npc1b*, *Npc2a*, and *Npc2b*, the putative sterol-*O*-acyltransferase gene *CG8112* and the intestinal SE hydrolase gene *magro*. We observed a 1.8-fold increase in *magro* mRNA levels in *Hsl[1]* mutants as compared to controls, whereas expression of all other sterol metabolic genes investigated remained largely unaltered in response to Hsl-deficiency (*Figure 3H*). Taken together, these data suggest that Hsl specifically regulates sterol homeostasis in adult *Drosophila* by promoting the catabolism of SE.

## Evolutionarily conserved adipocyte Hsl function controls organismal SE levels in *Drosophila*

To characterize a potential differential role of Hsl in the SE catabolism of specific tissues, we next measured SE in dissected body segments and organs of *Hsl[1]* mutant animals. In comparison to controls, SE levels were increased 2.6-, 2.8-, 2.4-, and 6.8-fold in *Hsl[1]* mutant heads, thoraces, abdomen, and carcasses, respectively. *Hsl[1]* mutant animals accumulated SE at lower levels also in the intestine but not in ovaries (*Figure 4A*). Moreover, *Hsl* deficiency did not provoke accumulation of LDs in the larval ring gland, a steroidogenic tissue with highly active sterol metabolism (*Figure 4B*). These findings imply an accumulation of SE in the fat body of *Hsl[1]* mutant animals as this tissue is present in all

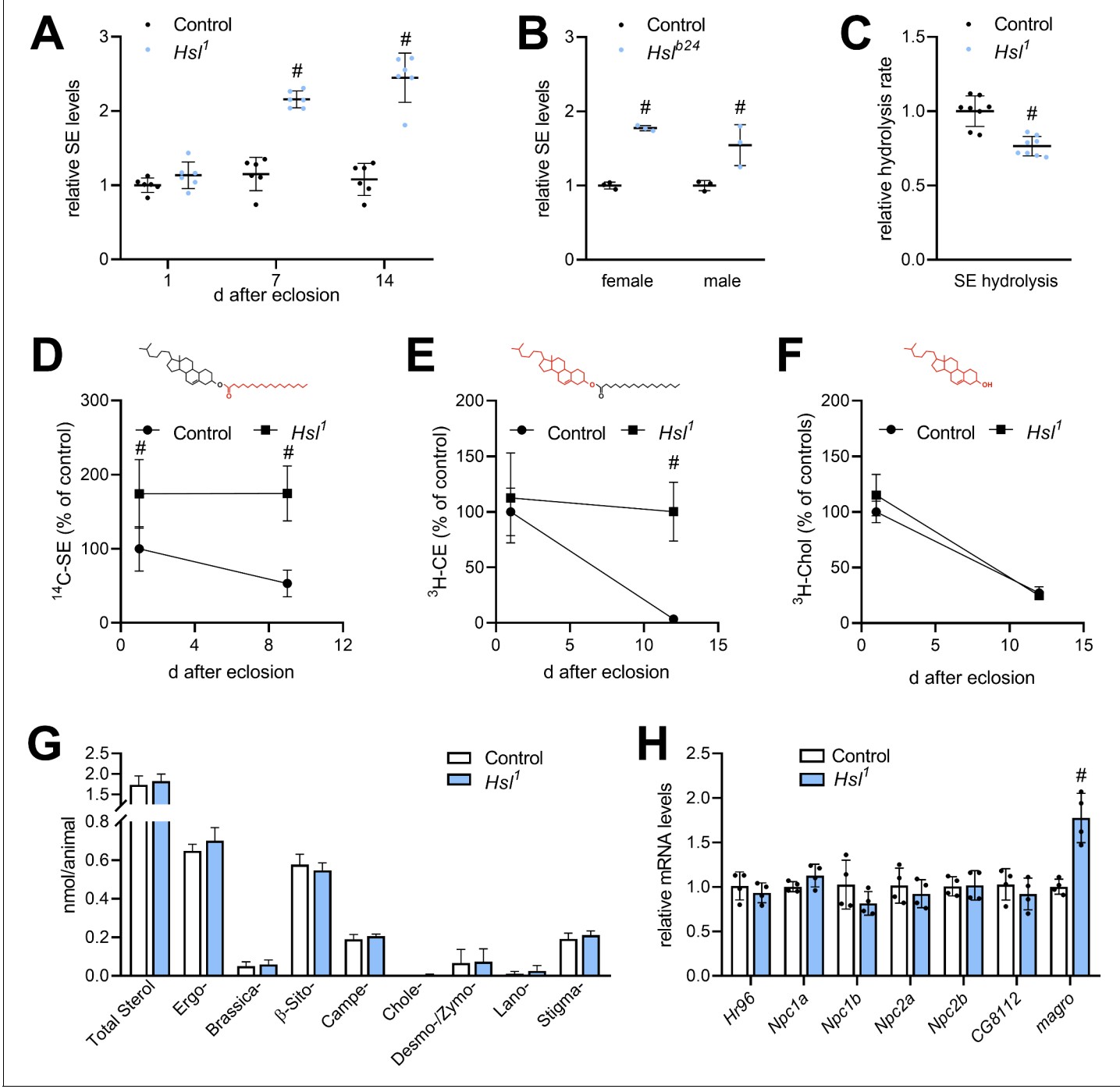

**Figure 3.** Sterol metabolism in *Hsl[1]* mutant flies. (A) SE levels in ad libitum fed control and *Hsl[1]* mutant flies at different times after eclosion. Data are normalized to 1-day-old (1d) control animals (n = 6). (B) SE levels in *Hsl[b24]* mutant males and females 7 days after eclosion. Data are normalized to controls (n = 3). (C) Normalized neutral SE hydrolase activities of *Hsl[1]* mutant and control abdominal extracts (n = 8). (D–F) Turnover of radiolabeled sterols in *Hsl[1]* mutant animals. Larvae were reared on food containing (D) $^{14}$C-FA or (E) $^{3}$H-cholesterol, switched to non-labeled food after eclosion and radioactivity in (D, E) SE, and (F) free sterol fractions was determined at the indicated timepoints (n = 4–5). Red and black colors in chemical structures indicate radiolabeled and unlabeled lipid moieties, respectively. (G) Non-esterified sterols in ad libitum fed control and *Hsl[1]* mutant animals as determined by shotgun MS (n = 7–8). (H) Relative mRNA levels of genes involved in sterol transport and metabolism. Relative mRNA levels were determined by qPCR and normalized to controls (n = 4). Data are presented as mean and SD. Statistical significance was determined by unpaired *t*-tests (#, p<0.05).

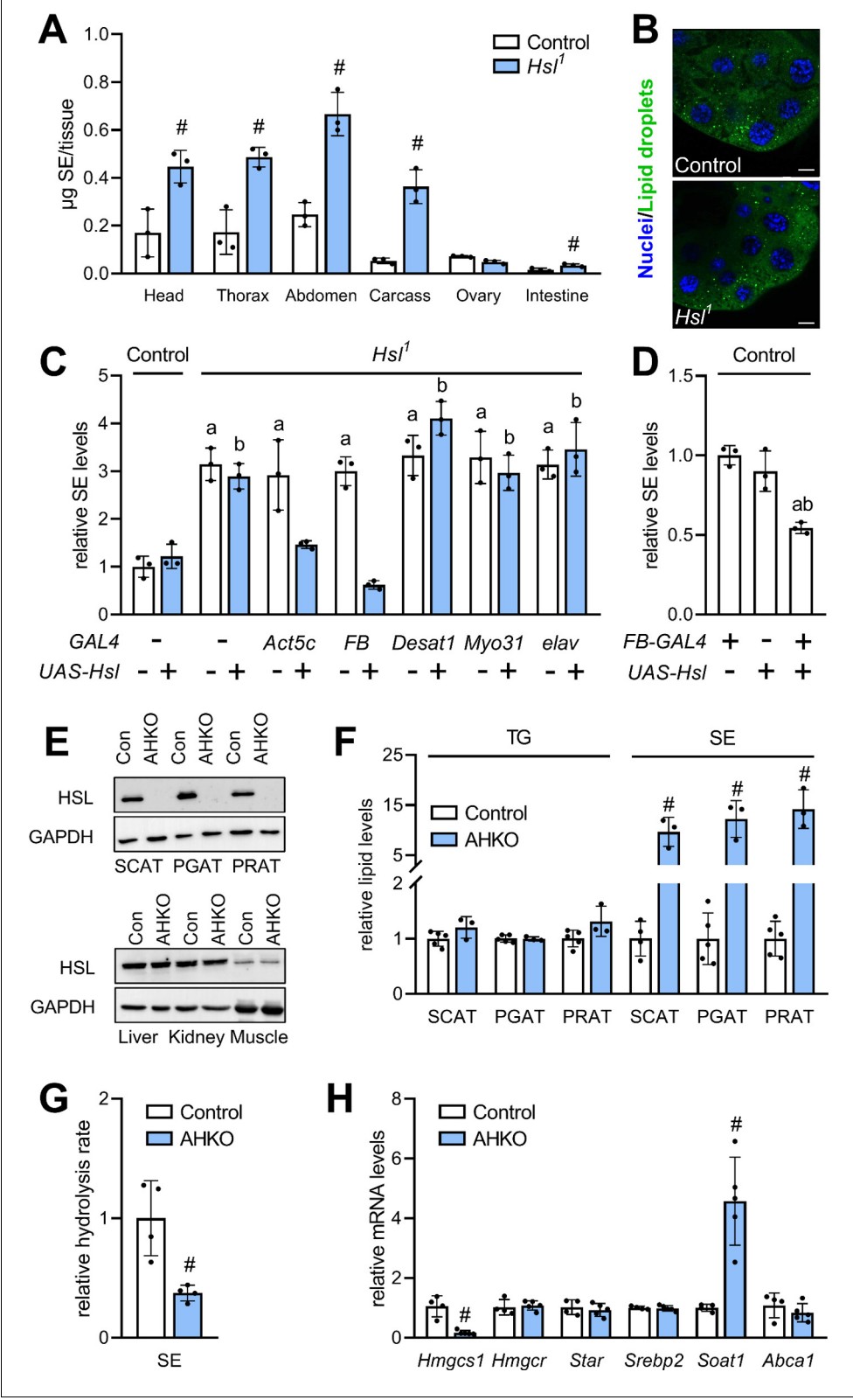

**Figure 4.** Adipocyte-autonomous control of organismal SE by Hsl. (**A**) Distribution of SE in body segments and tissues of adult *Hsl¹* mutant and control animals (n = 3). (**B**) Lipid droplets in steroidogenic ring glands of *Hsl¹* mutant and control prepupae. (**C**) Whole body SE after *GAL4*-driven re-expression of *UAS-Hsl* in all tissues (*Act5c*), fat body (*FB*), intestine (*Myo31DF*), oenocytes (*Desat1*), or nervous system (*elav*) of *Hsl¹* mutant animals. Levels are normalized to controls (n = 3). (**D**) Relative whole-body SE levels after fat body-specific expression of Hsl in control animals (n = 3). (**E**) Immunoblotting

*Figure 4 continued on next page*

Figure 4 continued

analysis of HSL in tissues of mice with adipocyte-specific disruption of the *HSL* gene (AHKO) and controls. GAPDH was used as a control. (F) Relative TG and SE levels in SCAT, PGAT, and PRAT of AHKO and control mice (n = 3–5). (G) SE hydrolase activities in soluble PGAT extracts of control and AHKO mice (n = 4). (H) Relative mRNA levels of sterol metabolism genes in PGAT of control and AHKO mice. mRNA levels were measured by qPCR and are normalized to controls (n = 4–5). Data are presented as means and SD. Statistical significance was determined by (A, F, G, H) unpaired *t*-tests (#, p<0.05) and (C, D) one-way ANOVA (a, p<0.05 compared to control; b, p<0.05 compared to control + *UAS-Hsl*).

body segments and enriched in carcass preparations. To more directly test this hypothesis, we restored *Hsl* expression in specific tissues of the *Hsl*[1] mutant animals by means of the UAS/Gal4 system. Ubiquitous or fat body-specific re-expression of *Hsl* by means of the *Act5c-GAL4* or *FB-GAL4* driver, respectively, rescued the SE overstorage phenotype of *Hsl*[1] mutant animals (*Figure 4C*). In contrast, neuronal, intestinal or oenocyte-specific re-expression of *Hsl* by means of *elav-GAL4*, *Myo31DF-GAL4* or *Desat1-GAL4* drivers, respectively, did not revert the SE overstorage phenotype of *Hsl*[1] mutants suggesting that Hsl acts autonomously in the fat body to control the organismal SE levels (*Figure 4C*). In line with this observation, ectopic expression of *Hsl* in the fat body of control flies decreased total body SE storage by 46% as compared to controls (*Figure 4D*). Since Hsl has been conserved during evolution, we next asked if an analogous pathway of sterol mobilization exists in mammalian adipocytes. To this end, we used mice with adipocyte-specific HSL-deficiency (AHKO) and analyzed storage lipid levels in the adipose tissue depots of these animals. As shown in *Figure 4E*, AHKO mice lack detectable HSL immunoreactivity in subcutaneous (SCAT), perigonadal (PGAT), and perirenal (PRAT) white adipose tissue depots but not in liver, kidney, or muscle. AHKO animals had similar amounts of TG as controls in their SCAT, PGAT, and PRAT but accumulated 10-, 12-, and 14-fold more SE, respectively, in these adipose tissue depots (*Figure 4F*). SE accumulation in AHKO adipose tissue depots was associated with a 63% decrease in neutral SE hydrolase activity as compared to controls (*Figure 4G*). To assess how disrupted SE hydrolysis affects sterol metabolic gene expression, we measured mRNA concentrations of genes involved in de novo synthesis, transport, and esterification of cholesterol. Notably, mRNA concentrations of the de novo cholesterol synthesis gene *Hmgcs1* were decreased by 83% in PGAT of AHKO mice as compared to controls arguing for a dysregulation of adipocyte sterol metabolic gene expression upon disrupted SE hydrolysis. Conversely, mRNA concentrations of the sterol-*O*-acyltransferase gene *Soat1* were increased by 4.8-fold, whereas *Hmgcr*, *Srebp2*, *Star*, *Abca1* were similarly expressed in both genotypes (*Figure 4H*). We conclude that Hsl acts as evolutionarily conserved adipocyte-autonomous regulator of SE catabolism and thereby controls whole-body SE levels in *Drosophila*.

## Maternal Hsl determines embryonic sterol homeostasis

*Hsl* is expressed throughout development with particularly high levels during early embryogenesis (*Bi et al., 2012*). To test if *Hsl* function in SE hydrolysis is relevant for the early *Drosophila* development, we compared embryonic free and esterified sterol levels of *Hsl*[1] mutant and control embryos. To discriminate between maternal and zygotic effects in embryos, we also included reciprocal crosses between control and *Hsl*[1] mutant parents. In controls, embryonic SE levels dropped by 66% within 6–8 hr after egg laying (AEL) and were almost undetectable at the end of embryogenesis (18–20 hr; *Figure 5A*). Embryos completely devoid of *Hsl* exhibited strongly delayed SE catabolism resulting in 1.6-, 4.0-, and 7.4-fold higher SE levels than controls at 0–2 hr, 6–8 hr, and 18–20 hr AEL, respectively. Heterozygous embryos from *Hsl*[1] mutant mothers and control fathers had 1.5-, 4.1-, and 4.1-fold higher SE levels than control embryos at 0–2 hr, 6–8 hr, and 18–20 hr AEL, respectively. In contrast, heterozygous embryos from *Hsl*[1] mutant fathers and control mothers exhibited similar SE catabolism rates than control embryos (*Figure 5A*) suggesting that maternal rather than zygotic Hsl determines embryonic SE breakdown. SE catabolism in control embryos correlated with an increase in free sterol content from 12.4 to 20.7 pmol per animal between 0–2 hr and 18–20 hr AEL. *Hsl*[1] mutant embryos had lower free sterol content than controls throughout embryogenesis and exhibited a blunted increase from 7.8 to 11.6 pmol per animal between 0–2 hr and 18–20 hr AEL (*Figure 5B*). This suggests that loss of Hsl causes impaired conversion of SE to free sterols during embryogenesis. Since embryonic SE catabolism was essentially dependent on the maternal genotype, we next asked if germline Hsl autonomously controls embryonic SE catabolism. To this end, we re-expressed *Hsl* specifically in the germline of mutant animals using the *nos-GAL4* driver. Enzyme

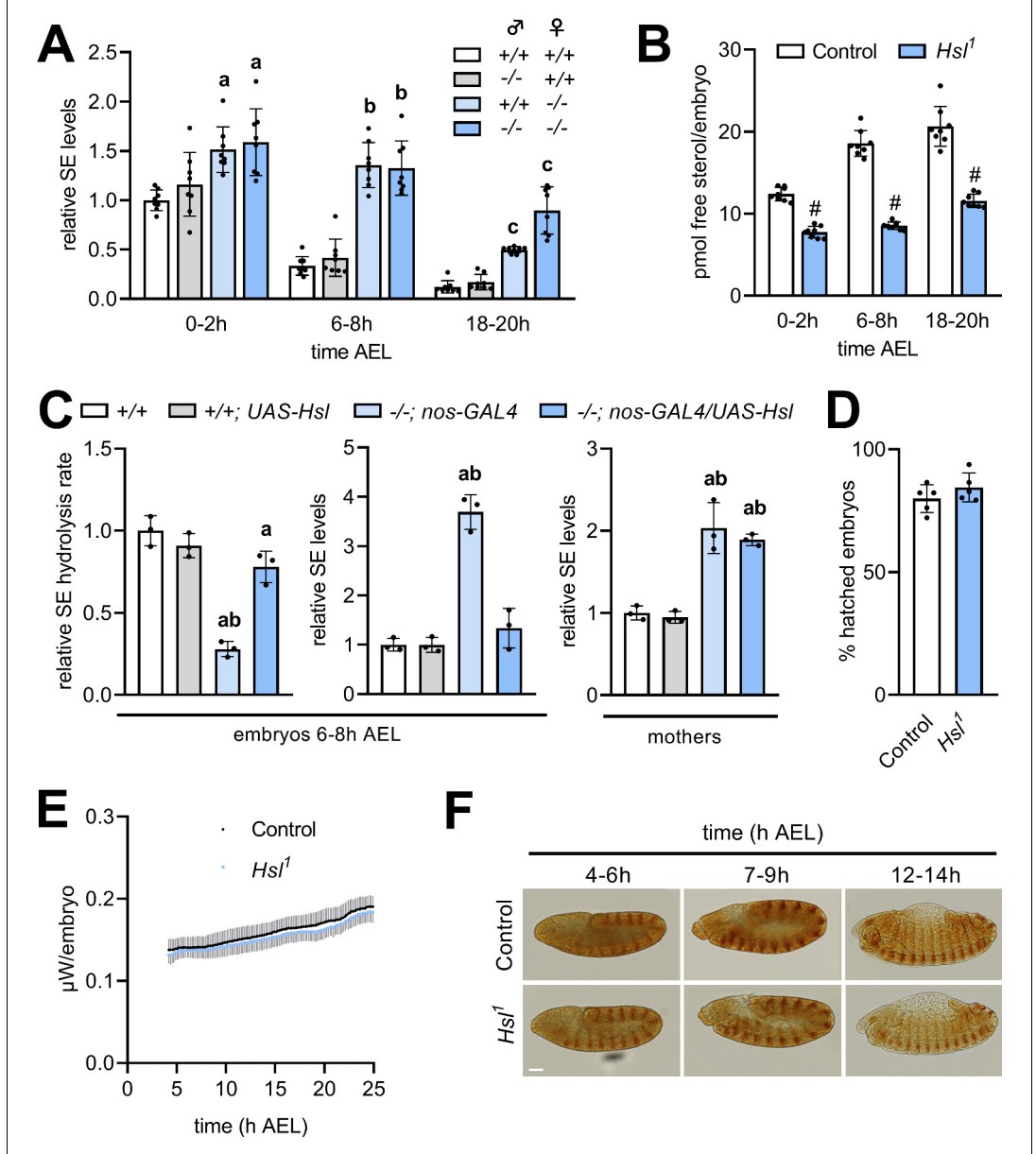

**Figure 5.** Maternal Hsl determines embryonic sterol homeostasis. (A) Relative SE levels in embryos derived from reciprocal matings between *Hsl1* (-/-) and control (+/+) animals at different times AEL (n = 8). (B) Free sterols in *Hsl1* and control embryos at different times AEL (n = 8). (C) Relative SE hydrolysis rates and SE levels in control and *Hsl1* mutant embryos and mothers after *nos-GAL4*-driven re-expression of *UAS-Hsl* in the germline (n = 3). (D) Hatching rates of control and *Hsl1* mutant embryos (n = 5). (E) Heat dissipation rates of control and *Hsl1* mutant embryos as determined by microcalorimetry (n = 12). (F) Immunohistochemical detection of Engrailed in control and *Hsl1* mutant embryos at different times AEL. Scale bar: 50 µm. Data are presented as means and SD. Statistical significance was determined by (B, D) unpaired *t*-tests (#, p<0.05), (A) one-way ANOVA (a, b, c, p<0.05 vs +/+ at 0–2 hr, 6–8 hr and 18–20 hr AEL, respectively) and (C) one-way ANOVA (a, p>0.05 vs +/+ and b, p>0.05 vs +/+; *UAS-Hsl*).

assays revealed a 72% decrease in embryonic SE hydrolysis rates of *Hsl1* as compared to control animals, which was largely reverted by re-expressing *Hsl* in the germline of mutant mothers (**Figure 5C**). This indicates that our strategy was successful in restoring active Hsl enzyme in mutant embryos. Germline-specific expression of *Hsl* fully prevented the SE accumulation in *Hsl1* mutant embryos but not in mothers consistent with a germline-autonomous role of the enzyme in SE catabolism (**Figure 5C**). To investigate if defective sterol homeostasis affects success rate or speed of the embryonic development of *Hsl1* mutants, we first measured hatching rates and found that under our

experimental conditions a similar fraction of control and *Hsl*[1] mutant embryos hatched into 1st instar larvae (*Figure 5D*). Between 4 hr and 24 hr AEL, thermal dissipation rates of control and *Hsl*[1] mutant embryos increased from 138 nW to 188 nW and from 132 nW to 182 nW, respectively, indicating similar metabolic rates in both genotypes (*Figure 5E*). Moreover, immunostainings of the segment-polarity protein Engrailed revealed that control and *Hsl*[1] mutant embryos reached comparable stages of embryogenesis at 4–6 hr, 7–9 hr, or 12–14 hr AEL (*Figure 5F*). Thus, although Hsl is rate-limiting for embryonic SE catabolism and required to adjust sterol levels during development, loss of Hsl is largely compatible with normal embryogenesis.

## Decreased sterol is compatible with the normal differentiation of the embryonic lipidome

Sterols are essential for the biophysical properties of biomembranes. To understand the significance of sterols in the context of other membrane lipids, we monitored the embryo lipidome in the early (0–2 hr AEL), middle (6–8 hr AEL), and late (18–20 hr AEL) phases of embryogenesis. In control animals, embryogenesis was associated with a gradual decrease in TG (by 12% and 58% at 6–8 hr and 18–20 hr AEL, respectively, compared to 0–2 hr AEL) but not DG (*Figure 6A*). This was accompanied by moderate increases in the levels of major glycerophospholipids like phosphatidylcholine (PC, 13% and 23% at 6–8 hr and 18–20 hr AEL, respectively), phosphatidylethanolamine (PE, 19% and 26% at 6–8 hr and 18–20 hr AEL, respectively), phosphatidylinositol (PI, 21% and 34% at 6–8 hr and 18–20 hr AEL, respectively), and phosphatidylserine (PS, 44% and 54% at 6–8 hr and 18–20 hr AEL, respectively; *Figure 6A*). Within this time frame, we observed even more pronounced increases in sphingolipids and ether-linked glycerophospholipids including ceramide (Cer, 79% and 75% at 6–8 hr and 18–20 hr AEL, respectively), ceramide phosphoethanolamine (CerPE, 59% and 114% at 6–8 hr and 18–20 hr AEL, respectively), and ether-linked PE (PE O-, 17% and 350% at 6–8 hr and 18–20 hr AEL, respectively; *Figure 6A*). An exception to this general trend was phosphatidylglycerol (PG), which exhibited a subtle decrease during embryogenesis (3% and 7% at 6–8 hr and 18–20 hr AEL, respectively; *Figure 6A*). The net increase in membrane lipid content was associated with distinct shifts in the molecular composition of each lipid class. PC, PE, PI, and PS shifted toward increased acyl chain length at 18–20 hr AEL when compared to 0–2 hr AEL (*Figure 6B*). A similar shift was observed for Cer, CerPE, and PE O-, whereas PG shifted toward decreased acyl chain length (*Figure 6—figure supplement 1A*). Moreover, PC, PI, Cer, CerPE, and PE O- pools remodeled toward decreased acyl chain saturation whereas PE, PS, and PG became slightly more saturated (*Figure 6—figure supplement 1A*). Although these general shifts in total level, acyl chain length, and saturation degree of each lipid class were often moderate, distinct lipid species including many polyunsaturated glycerophospholipids exhibited more dynamic changes. Examples include PC 36:4, PE 36:4, and PI 36:4, which increased by 169%, 125%, and 194%, respectively, between early and late embryogenesis (*Figure 6C*, *Supplementary file 1d*). However, it should be noted that these species were low abundant and therefore contributed little to the overall saturation degree of each lipid class. When compared with control animals, *Hsl*[1] mutants exhibited similar decreases in TG and increases in membrane lipid levels during embryogenesis (*Figure 6A,B*; *Figure 6—figure supplement 1*; *Supplementary file 1*). Total PC increased slightly less in *Hsl*[1] compared to control animals between early and late embryogenesis (16% vs 23%), which was largely due to lower amounts of medium-chain PC species containing 30 or less carbon atoms (*Figure 6C*; *Supplementary file 1*). Instead, increases in PC species with higher chain length and more double bonds were comparable between *Hsl*[1] and control embryos (*Figure 6C*, Supplementary file 1). Likewise, the molecular composition of other phospholipids and ceramides changed with similar dynamics in control and *Hsl*[1] mutant animals (*Figure 6B,C*; *Figure 6—figure supplement 1*; *Supplementary file 1*). Taken together, these data suggest that (1) Hsl acts as specific regulator of sterols but not of other lipids during embryogenesis and (2) decreased sterols as a consequence of defective SE catabolism are largely compatible with normal remodeling of membrane lipids in *Hsl*[1] mutant embryos.

## Hsl controls maternal sterol transport to the oocyte and improves fecundity

In order to more comprehensively understand the consequences of defective sterol handling for reproductive success we analyzed the quantity and quality of eggs laid by control and *Hsl*[1] mutant

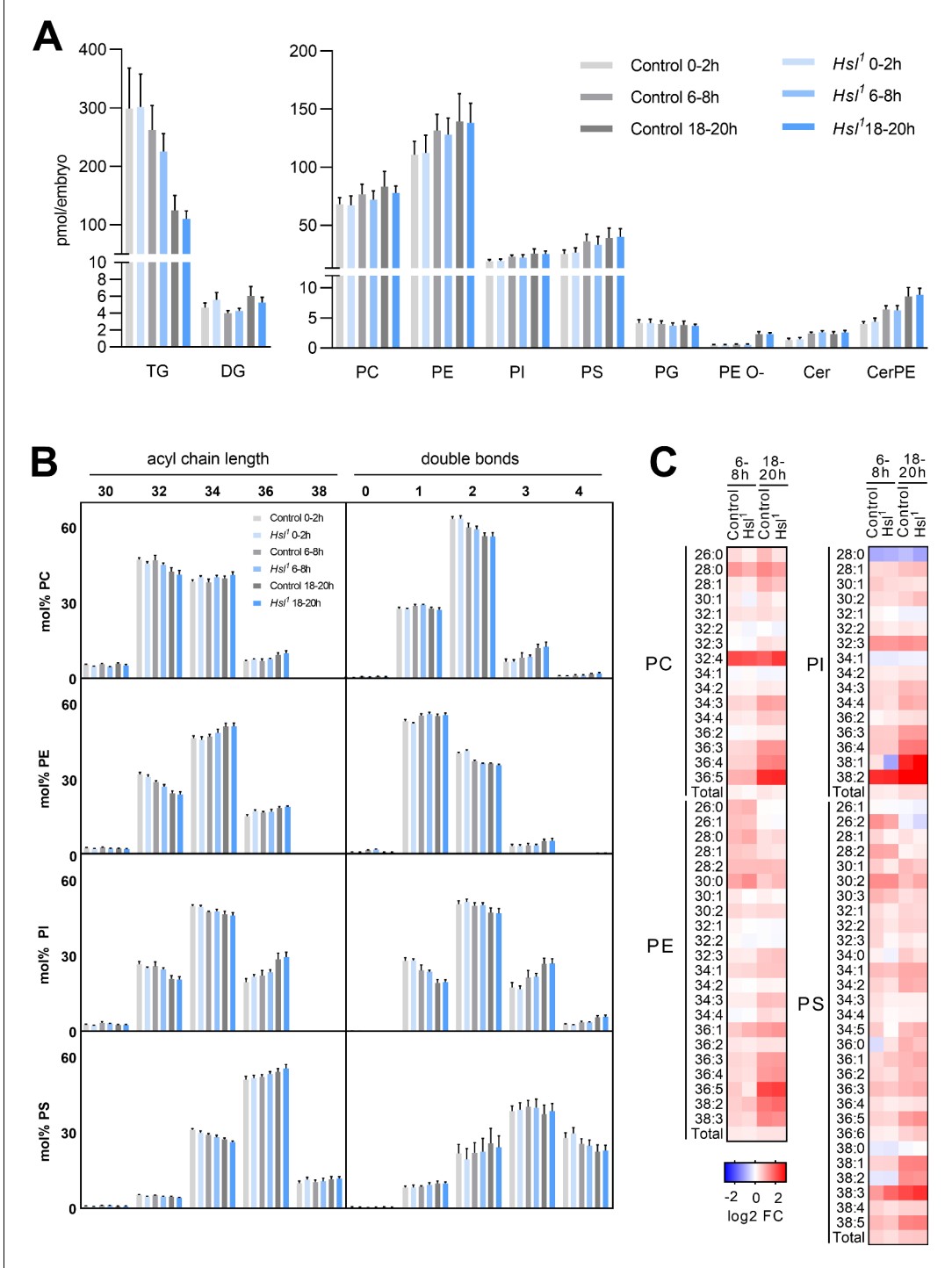

**Figure 6.** Differentiation of the embryo lipidome in control and $Hsl^1$ mutants. (**A**) Total levels of neutral (left panel) and polar (right panel) lipid classes during embryogenesis of control and $Hsl^1$ animals (n = 8). (**B**) Combined acyl chain lengths and double bonds in major glycerophospholipid classes of control and $Hsl^1$ mutant embryos at different timepoints during embryogenesis (n = 8). (**C**) Relative changes of individual glycerophospholipid species in control and $Hsl^1$ animals between early (0–2 hr AEL) and mid (6–8 hr AEL) or late (18–20 hr AEL) embryogenesis (n = 8). Data are presented as (**A, B**) means and SD or (**C**) log2-transformed fold changes (FC) normalized to early (0–2 hr AEL) control embryos.

The online version of this article includes the following figure supplement(s) for figure 6:

**Figure supplement 1.** Changes in the lipidome during embryogenesis in control and $Hsl^1$ mutants.

females. We used a lipid-depleted medium (LDM), which is naturally low in sterols and other lipids, to increase the reliance of the animals on endogenous sterol reserves and counted eggs and 1st instar larvae produced by individual females. *Hsl*[1] females had 24% and 20% less cumulative egg output than control females after 4 days and 7 days on LDM, respectively (*Figure 7A*). In both genotypes, the dietary switch to LDM was associated with a drastic time-dependent decline in egg hatchability reaching a plateau after 7 days. *Hsl*[1] females produced 37% and 25% less viable progeny than control females within 4 days and 7 days on LDM, respectively, indicating slightly higher developmental success of control embryos in the early period on LDM (*Figure 7A*). Indeed, hatching rates of *Hsl*[1] mutant animals were 14% lower after 4 days on LDM as compared to controls (*Figure 7—figure supplement 1*). After dietary supplementation with exogenous cholesterol, *Hsl*[1] females showed 16% and 14% lower egg output than control females at 4 days and 7 days, respectively, suggesting that dietary cholesterol partially rescued the fecundity defect of *Hsl*[1] mutant animals (*Figure 7B*). Cholesterol addition to LDM rescued the decline in embryo hatchability in both genotypes. As a consequence of similar hatching rates in both genotypes, *Hsl*[1] females produced 17% and 14% less viable offspring than control females at 4 days and 7 days, respectively (*Figure 7B*, *Figure 7—figure supplement 1*). To understand how defective SE catabolism in the maternal organism affects fecundity, we next assessed lipid transfer to control and *Hsl*[1] mutant oocytes. Vitellogenic follicles of control but also *Hsl*[1] females produced numerous LDs at stage 10 of oogenesis suggesting that oocyte lipid loading is generally functional in *Hsl*[1] mutant animals (*Figure 7C*). To more specifically monitor sterol transport, we followed the incorporation of labeled cholesterol into embryos produced by mothers of both genotypes and found that *Hsl*[1] mutants transferred 20% and 24% less labeled cholesterol into the embryos laid at 3 days and 4 days after female mating (*Figure 7D*). This suggests that Hsl indeed promotes the mobilization and transfer of maternal sterols to the progeny. Since Hsl contributes to SE catabolism in both, maternal fat body and germline, we finally used genetic approaches to address nonautonomous contributions of maternal fat body Hsl to fecundity. Fat body-specific knockdown of *Hsl* expression by means of a *Cg-GAL4* driven *Hsl shRNA* transgene reduced cumulative egg and 1st instar numbers by 27–29% and 32–34%, respectively, after 4 days on LDM as compared to controls harboring either the *Hsl shRNA* or the *Cg-GAL4* only (*Figure 7E*). Similar to complete Hsl-deficiency the fecundity defects induced by fatbody-specific Hsl knockdown were less pronounced after 7 days on LDM (25–27% less egg numbers, 14–24% less 1st instar larvae compared to controls, *Figure 7E*). In a complementary approach, we re-expressed *Hsl* in the fat body of the *Hsl*[1] mutant animals via the *FB-GAL4* driver and assessed fecundity on LDM. After 4 days on LDM, *Hsl*[1] mutant animals harboring either the *UAS-Hsl* transgene or the *FB-GAL4* element produced significantly less eggs (21–29%) and 1st instar larvae (24–29%) than controls with the *FB-GAL4* element (*Figure 7F*). Fat body-specific *Hsl* re-expression was sufficient to ameliorate fecundity defects associated with complete Hsl-deficiency (14% less eggs and 8% less 1st instar larvae compared to controls at 4 days on LDM, *Figure 7F*). Taken together, these data indicate that optimal egg production and reproductive success requires Hsl-mediated SE catabolism in the fat body of the mothers, particularly when dietary sterol is limited.

## Discussion

Cycles of storage lipid build-up and breakdown are an integral part of lipid homeostasis and orchestrate the distribution of lipid resources between tissues, ontogenetic stages and generations. In this study, we present the first in vitro and in vivo analysis to identify Hsl as major SE hydrolase in *Drosophila*. We complement these fly data with the analysis of adipocyte sterol metabolism using tissue-specific knockout mice to suggest an ancestral function of Hsl-related enzymes in sterol homeostasis of animals. In flies, this enzymatic activity is of physiological relevance since we demonstrate an intergenerational role of Hsl in sterol metabolism of the progeny.

Hsl belongs to a phylogenetically ancient family of lipid hydrolases implicated mainly in TG mobilization (*Bi et al., 2012*; *Deb et al., 2006*; *Langin and Holm, 1993*; *Liu et al., 2017*). In mammalian adipocytes, Hsl acts as principal TG and DG hydrolase in parallel and downstream of ATGL (*Haemmerle et al., 2002a*; *Zhang et al., 2019*). Accordingly, Hsl-deficient mice cannot adequately increase adipose tissue TG lipolysis in response to starvation or exercise (*Fernandez et al., 2008*; *Haemmerle et al., 2002b*). The loss of adipose tissue Hsl culminates in progressive lipodystrophy, liver steatosis, and insulin resistance (*Xia et al., 2017*). A critical step in Hsl activation occurs post-

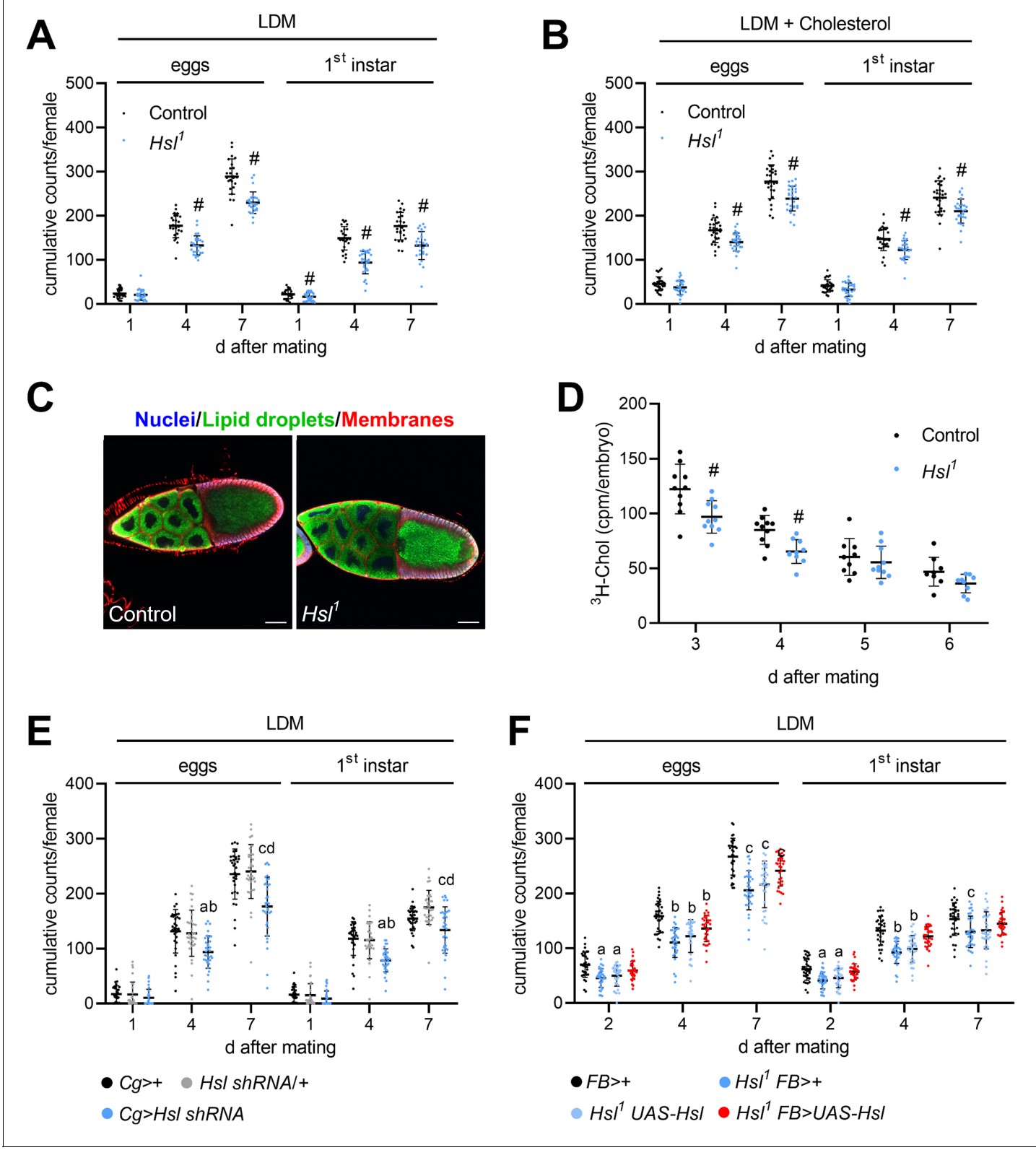

**Figure 7.** Impaired fecundity and egg loading in *Hsl[1]* mutant flies. (**A, B**) Cumulative numbers of eggs and hatched 1st instar larvae per female on (**A**) LDM (n = 25–28) or (**B**) LDM with 0.01% cholesterol (n = 29). Note that LDM contains only traces of sterols and other lipids. (**C**) Lipid loading in control and *Hsl[1]* mutant oocytes. Stage 10 follicles were stained with Hoechst 33342, BODIPY 493/503, and Cellmask Deep Red to detect nuclei, lipid droplets, and cell membranes, respectively, and imaged by confocal fluorescence microscopy. Scale bars: 50 µm. (**D**) Transfer of [3]H-cholesterol into control and

*Figure 7 continued on next page*

*Figure 7 continued*

*Hsl*[1] mutant embryos collected at different times after mating (n = 9–10). (**E, F**) Cumulative numbers of eggs and hatched 1st instar larvae per female upon (**E**) fat-body-specific RNAi-mediated downregulation of *Hsl* expression by means of a *Cg-GAL4*-driven *Hsl* shRNA transgene (n = 32–34) and (**F**) fat-body-specific rescue of *Hsl* expression in *Hsl*[1] mutants by means of a *FB-GAL4*-driven *UAS-Hsl* transgene (n = 33–34). Data are presented as means and SD. Statistical difference was determined by (**A,B,D**) unpaired *t*-tests (#, $p<0.05$ vs control), (**E**) one-way ANOVA (a, c, $p<0.05$ vs *Cg>*+ at 4 days and 7 days after mating; b, d, $p<0.05$ vs *Hsl* shRNA*/+* at 4 days and 7 days after mating) and (**F**) one-way ANOVA (a,b,c $p>0.05$ vs *FB>*+ at 2 days, 4 days, and 7 days after mating, respectively).

The online version of this article includes the following figure supplement(s) for figure 7:

**Figure supplement 1.** Hatching rates of control and *Hsl*[1] mutant embryos.

translationally by signaling events induced by catecholamines and glucagon (*Lass et al., 2011*). These catabolic hormones augment adipocyte lipolysis in response to starvation or exercise via the concerted activation of ATGL and Hsl (*Lass et al., 2011*). A similar molecular system operates in the *Drosophila* fat body, which relies on Bmm/ATGL and Akh/Glucagon-activated lipase(s) for full lipolytic output (*Grönke et al., 2007*). The presence of a *Drosophila* Hsl ortholog has fostered the hypothesis that this enzyme acts as major effector of Akh in TG lipolysis (*Lee et al., 2013*; *Lehmann, 2018*; *Zheng et al., 2016*). However, two observations in our study argue against this notion. First, Hsl-deficient flies do not accumulate excessive TG, which is a common biochemical manifestation of defective TG lipolysis and a signature of *Akh* or *bmm* loss-of-function mutants (*Gáliková et al., 2015*; *Grönke et al., 2005*; *Hofbauer et al., 2020*). Second, unlike *Akh bmm* double-deficient mutants, combined deficiencies of *Hsl* with *bmm* or *Akh* do not abrogate TG mobilization (*Grönke et al., 2007*). Furthermore, the energy homeostasis profile of *Hsl*[1] mutants is remarkably normal with the exception of a mildly elevated glycogen content of unknown etiology, which deserves future research attention. Taken together, these findings provide unequivocal evidence for the existence of Hsl/Bmm-unrelated fat body TG lipase(s) in *Drosophila*. Biochemical studies in *Manduca sexta* implicated a TG lipase with homology to *Drosophila* PAPLA1 in Akh-mediated TG lipolysis (*Arrese et al., 2006*). However, similar to *Hsl*[1] mutants, *PAPLA1*-deficiency is compatible with normal starvation-induced TG catabolism (*Gáliková et al., 2017*). It remains possible that several enzymes including Hsl and PAPLA1 act in redundancy to execute Akh-induced TG lipolysis. In this scenario, genetic deficiencies of either enzyme would be compensated by the other. The relative significance of each lipase for TG lipolysis may also depend on specific environmental conditions or developmental stages. Previous studies identified Hsl and Lip3 as regulators of 3rd instar larval acylglycerol metabolism (*Bi et al., 2012*; *Bülow et al., 2018*). Together with our data this observation suggests that different ontogenetic stages use specific enzyme sets to execute TG lipolysis in *Drosophila*.

Our study supports a major and possibly ancestral function of Hsl in the degradation of SE. Sterol biosynthesis is an evolutionary invention of early eukaryotes but has been secondarily lost in various metazoan lineages including insects (*Babin and Gibbons, 2009*). Accordingly, *Drosophila* is a sterol auxotroph, which requires robust molecular mechanisms for effective uptake and distribution of dietary sterols (*Carvalho et al., 2010*). The *Drosophila* midgut plays a central role in the perception, metabolism, and distribution of diet-derived sterols. In this tissue, the nuclear receptor Hr96 acts as a sterol sensor and orchestrates transcriptional responses to fluctuations in sterol supply (*Bujold et al., 2010*; *Horner et al., 2009*; *Sieber and Thummel, 2012*). Hr96 facilitates SE catabolism by promoting expression of the SE/TG lipase Magro. Consequently, the loss of Hr96 or Magro leads to an accumulation of SE (*Sieber and Thummel, 2012*). Although this phenotype resembles Hsl-deficient animals, it most likely represents a selective disruption of SE catabolism in enterocytes. Conversely, our genetic analysis implies that Hsl acts mainly in fat body SE catabolism. This suggests the presence of tissue-autonomous SE pools in *Drosophila* that are controlled by specific subsets of enzymes. Substantial residual neutral SE hydrolase activities in *Hsl* mutant abdominal extracts argue for the presence of additional, yet unidentified, SE hydrolases in *Drosophila*. It is currently unclear, how these different enzymatic systems interact to coordinate organismal sterol homeostasis. However, increased concentrations of *magro* mRNA in Hsl-deficient animals indicate a crosstalk between different enzymes and tissues in the control of SE homeostasis.

Defective SE degradation in Hsl mutants correlates with reduced sterol transport into embryos. This indicates a preferred use of the maternal SE pool for vitellogenesis. *Drosophila* tissues acquire

specific sterol patterns during development and exchange their sterol pools at different rates (*Carvalho et al., 2012*; *Knittelfelder et al., 2020*). Thus, specific transport routes may regulate the tissue distribution of sterols. Since the majority of the *Drosophila* SE pool resides in the fat body, it is reasonable to assume that Hsl is part of a specific sterol transport route between adipocytes and vitellogenic follicles. Modulation of Hsl activity may thus be a mechanism of channeling sterols selectively toward ovaries. The loading of oocytes with lipids is essential for the survival of the offspring (*Parra-Peralbo and Culi, 2011*). It is regulated by a hierarchical system in which local ecdysone signaling promotes expression of ovarian lipophorin receptors via the transcription factor SREBP (*Sieber and Spradling, 2015*). Similar hormonal signals may also act on the fat body to communicate increased ovarian demand for lipid resources. Although egg production is tightly coupled to dietary nutrient supply, the fat body may act as an additional nutriostat for vitellogenesis. Lower egg laying capacity has been observed in genetically lean flies that fail to build up or maintain TG reserves (*Sieber and Spradling, 2015*). Analogously, reduced egg laying in mutant females lacking adipocyte *Hsl* function provides evidence that egg production also depends on maternal SE reserves. During oogenesis, the maternal germline provides the embryos with Hsl function for early degradation of embryo SE and further amplification of embryonic free sterol levels. This observation illustrates an intergenerational coupling of sterol metabolism via temporally and spatially shifted waves of storage lipid breakdown.

Our analyses unravel dynamic changes in the membrane lipidome during *Drosophila* embryogenesis. Quantitative increases in most membrane lipid classes, especially sphingolipids and ether-linked glycerophospholipids, coincide with enrichments in polyunsaturated FA species with increased acyl chain length. These changes likely reflect maturation of biomembranes during cell differentiation and organogenesis. Hsl specifically contributes to this developmental maturation of the embryo lipidome by increasing free sterols through early SE hydrolysis. The formation of sterol-rich liquid crystalline microdomains is considered essential for distinct membrane functions such as signaling and trafficking (*Lingwood and Simons, 2010*). Consequently, membrane sterol content is tightly regulated in many eukaryotic species (*Luo et al., 2020*). Surprisingly, *Hsl* mutants develop with normal success and speed despite lacking 50% of free sterol at the end of embryogenesis. Marginal differences between control and mutant lipidomes indicate a previously unexpected tolerance of the membrane to substantial fluctuations in sterol content and challenge the dogma that membrane sterol concentrations have to be maintained within a narrow range to allow for proper cellular function. Similarly, substantial variations in tissue sterol levels have also been reported in *Drosophila* larvae raised on different diets indicating that *Drosophila* biomembranes may accommodate variable sterol levels without overt dysfunction (*Carvalho et al., 2010*). Although the sterol surplus provided by Hsl is not essential under artificial laboratory conditions, it will be interesting to learn whether Hsl function provides a fitness advantage under the more variable conditions of a natural environment.

Remarkably, SE accumulation is also a hallmark of mammalian Hsl-deficient adipocytes arguing for a conserved role of this enzyme in adipocyte sterol homeostasis. Mammalian Hsl is an established regulator of SE breakdown in multiple tissues including testis, adrenal tissue, intestine, and liver (*Kraemer et al., 2004*; *Obrowsky et al., 2012*; *Osuga et al., 2000*; *Sekiya et al., 2008*). While a function of Hsl in adipocyte SE breakdown has been previously suggested by in vitro assays and altered gene expression signatures, it has not yet been substantiated by correlating it with adipocyte SE levels (*Harada et al., 2003*; *Osuga et al., 2000*). Mammalian adipose tissue is an important sterol sink as it contains between 25% and 50% of total body cholesterol (*Krause and Hartman, 1984*). Adipocytes also actively participate in reverse cholesterol transport (*Prattes et al., 2000*; *Zhang et al., 2010*). However, the majority of adipocyte sterol is non-esterified due to low sterol-*O*-acyltransferase activities in this tissue. Thus, the relevance of adipocyte SE as intermediate for systemic sterol transport in mammals is unclear. SE have recently been implicated in adipocyte differentiation and the formation of functional microdomains at LDs suggesting that SE function in subcellular rather than systemic sterol homeostasis (*Hansen et al., 2017*; *Xu et al., 2019*; *Zhu et al., 2018*). Perturbed expression of *Hmgcs1* and *Soat1* mRNA in AHKO adipose tissue likewise argues in favor of a Hsl function in regulating local, tissue-autonomous sterol homeostasis. If and how these alterations in SE storage and sterol metabolic gene expression translate into altered cellular functions remains to be investigated. It is well established that hepatocytes rather than adipocytes act as major hub of cholesterol distribution in mammals (*Yu et al., 2019*). Intriguingly, previous studies implicated mammalian Hsl in hepatic SE degradation (*Sekiya et al., 2008*). Thus, *Drosophila* fat

body may integrate liver- and adipose tissue-like functions as both, sterol sink and sterol transport center with Hsl contributing to the latter function. While our data indicate that adipocyte Hsl guarantees efficient lipid allocation of the *Drosophila* progeny a similar function of mammalian Hsl has yet to be established. Male Hsl mutant mice are infertile due to azoo- or oligospermia (*Osuga et al., 2000*). However, this pathology and the accompanying SE accumulation result from a testis-autonomous manifestation of Hsl-deficiency and are unrelated to the adipose tissue phenotype of Hsl-deficient mice (*Fortier et al., 2005*; *Vallet-Erdtmann et al., 2004*). Although Hsl-deficient female mice are not grossly infertile maternal reproductive health and offspring development have not been systematically analyzed in Hsl-deficient mice (*Chung et al., 2001*). Our study of *Drosophila* Hsl indicates an important role of this enzyme in the crosstalk between adipose tissue lipolysis, reproduction, and development. In light of the remarkable functional conservation of adipocyte Hsl during evolution, we believe that our study exposes an exciting future avenue for lipid and developmental research on mammalian and invertebrate model systems.

# Materials and methods

## Key resources table

| Reagent type (species) or resource | Designation | Source or reference | Identifiers | Additional information |
|---|---|---|---|---|
| Strain, strain background (*Drosophila melanogaster*) | $w^{1118}$ | Vienna *Drosophila* Research Center (VDRC) | Cat#: 60000 | |
| Genetic reagent (*Drosophila melanogaster*) | $Hsl^1$ | This study | | See Materials and methods |
| Genetic reagent (*Drosophila melanogaster*) | UAS-Hsl | This study | | See Materials and methods |
| Genetic reagent (*Drosophila melanogaster*) | UAS-Hsl-EGFP | This study | | See Materials and methods |
| Genetic reagent (*Drosophila melanogaster*) | $bmm^1$ | *Grönke et al., 2005* doi: 10.1016/j.cmet.2005.04.003 | FLYB: FBal0195572 | |
| Genetic reagent (*Drosophila melanogaster*) | $Akh^A$ | *Gáliková et al., 2015* doi: 10.1534/genetics.115.178897 | FLYB: FBal0319563 | |
| Genetic reagent (*Drosophila melanogaster*) | FB-GAL4 | *Grönke et al., 2003* doi: 10.1016/s0960-9822(03)00175–1 | | |
| Genetic reagent (*Drosophila melanogaster*) | Act5c-GAL4 | Bloomington *Drosophila* Stock Center (BDSC) | Cat#: 4414 FLYB: FBst0004414 | backcrossed to a $w^{1118}$ strain |
| Genetic reagent (*Drosophila melanogaster*) | Cg-GAL4 | Bloomington *Drosophila* Stock Center (BDSC) | Cat#: 7011 FLYB: FBst0007011 | |
| Genetic reagent (*Drosophila melanogaster*) | nos-GAL4 | Bloomington *Drosophila* Stock Center (BDSC) | Cat#: 64277 FLYB: FBst0064277 | |
| Genetic reagent (*Drosophila melanogaster*) | Hsl shRNA | Bloomington *Drosophila* Stock Center (BDSC) | Cat#: 65148 FLYB: FBst0065148 | |
| Cell line (*Chlorocebus aethiops*) | COS-7 | American Type Culture Collection (ATCC) | Cat#: CRL-1651 RRID: CVCL_0224 | |
| Antibody | Anti-Mouse IgG-Horseradish Peroxidase antibody; sheep | GE Healthcare | Cat#:NA931 RRID: AB_772210 | WB (1:5,000) |
| Antibody | Anti-Rabbit IgG (H+L)-Horseradish Peroxidase antibody; goat polyclonal | Vector Laboratories | Cat#: PI-1000 RRID: AB_2336198 | WB (1:5,000) |
| Antibody | Anti-engrailed/invected antibody; mouse monoclonal | Developmental Studies Hybridoma Bank (DSHB) | Cat#: 4D9 RRID: AB_528224 | IHC (1:100) |
| Antibody | HSL antibody; rabbit polyclonal | Cell Signaling Technology | Cat#: 4107 RRID: AB_2296900 | WB (1:1,000) |

*Continued on next page*

*Continued*

| Reagent type (species) or resource | Designation | Source or reference | Identifiers | Additional information |
|---|---|---|---|---|
| Antibody | Rabbit Anti-GAPDH antibody, unconjugated, clone 14C10; rabbit monoclonal | Cell Signaling Technology | Cat#: 2118 RRID: AB_561053 | WB (1:10,000) |
| Antibody | Anti-His antibody, unconjugated; mouse monoclonal | GE Healthcare | Cat#: 27-4710-01 RRID: AB_771435 | WB 1:5000 |
| Recombinant DNA reagent | pFLC-1 [*Hsl*] | Drosophila Genomics Resource Center (DGRC) | Cat#: RE52776 | Used for amplification of *Hsl* cDNA |
| Transfected construct (*D. melanogaster*) | pcDNA4/Hismax C [*His_6-Hsl*] | This study | | See Materials and methods |
| Recombinant DNA reagent | pUAST [*Hsl*] | This study | | See Materials and methods |
| Recombinant DNA reagent | pUAST [*Hsl-EGFP*] | This study | | See Materials and methods |
| Sequence-based reagent | Oligonucleotides, Primers | Thermo Fisher Scientific Invitrogen | | See Materials and methods |
| Commercial assay or kit | Triglycerides reagent | Thermo Fisher Scientific | Cat#: 981786 | |
| Commercial assay or kit | NEFA-HR(2) | Fujifilm Wako Diagnostics | Cat#: 999–34691, Cat#: 995–34791 Cat#: 991–34891 Cat#: 993–35191 Cat#: 276–76491 | |
| Commercial assay or kit | Glucose (GO) Assay kit | Sigma Aldrich | Cat#: GAGO20-1KT | |
| Commercial assay or kit | RNeasy Mini kit | QIAGEN | Cat#: 74104 | |
| Commercial assay or kit | QIAGEN Plasmid Midi Kit | QIAGEN | Cat#: 12143 | |
| Commercial assay or kit | TRIzol | Thermo Fisher Scientific Invitrogen | Cat#: 15596026 | |
| Commercial assay or kit | SuperScript III First-Strand Synthesis Supermix | Thermo Fischer Scientific Invitrogen | Cat#: 18080051 | |
| Commercial assay or kit | QuantiTect Reverse Transcription Kit | QIAGEN | Cat#: 205313 | |
| Commercial assay or kit | iTaq Universal SYBR Green Supermix | Bio-Rad | Cat#: 1725120 | |
| Chemical compound, drug | BODIPY 493/503 | Thermo Fisher Scientific Invitrogen | Cat#: B2103 | 5 µg/ml |
| Chemical compound, drug | Hoechst 33342 | Sigma Aldrich | Cat#: 14533 | 5 µg/ml |
| Chemical compound, drug | CellMask Deep Red | Thermo Fisher Scientific Life Technologies | Cat#: C10046 | 1 µg/ml |
| Chemical compound, drug | LipidTOX Deep Red | Thermo Fisher Scientific Life Technologies | Cat#: H3477 | (1:1,000) |
| Chemical compound, drug | Lipid standards | Avanti Polar Lipids Sigma Aldrich Toronto Research Chemicals | | See Materials and methods |
| Chemical compound, drug | Cholesterol | Sigma Aldrich | Cat#: C3045 | |
| Chemical compound, drug | Cholesterol [25,26–3H] | American Radiolabeled Chemicals | Cat#: ART-1987 | |

*Continued on next page*

*Continued*

| Reagent type (species) or resource | Designation | Source or reference | Identifiers | Additional information |
|---|---|---|---|---|
| Software, algorithm | LipidXplorer 1.2.7 | *Herzog et al., 2012* doi:10.1371/journal.pone.0029851 | | |
| Software, algorithm | Lipid Data Analyzer | *Hartler et al., 2011* doi:10.1093/bioinformatics/btq699 | | |

## Chemicals and reagents

An ECL Anti-Mouse IgG-Horseradish Peroxidase antibody from sheep s (Cat#: NA931; RRID: AB_772210) and a monoclonal Anti-His antibody from mouse (Cat#: 27-4710-01; RRID: AB_771435) were purchased from GE Healthcare (Chicago, IL). A monoclonal Anti-engrailed/invected antibody from mouse (Cat#: 4D9; AB_528224) was obtained from Developmental Studies Hybridoma Bank (Iowa City, IA). A polyclonal HSL antibody from rabbit (Cat#: 4107; RRID: AB_2296900) and a monoclonal Anti-GAPDH antibody from rabbit (Cat#: 2118, clone 14C10; RRID: AB_561053) were obtained from Cell Signaling Technology (Cambridge, UK). A polyclonal Anti-Rabbit IgG (H+L)-Horseradish Peroxidase antibody from goat (Cat#: PI-1000; RRID: AB_2336198) was purchased from Vector Laboratories (Peterborough, UK). TG 54:3 (triolein/trioleoylglycerol; Cat#: T7140), DG 36:2 (1,2-dioleoyl-*rac*-glycerol; Cat#: D8397), MG 18:1 (1-oleoyl-*rac*-glycerol; Cat#: M7765), egg yolk phosphatidylcholine (Cat#: P3556), soybean phosphatidylinositol (Cat#: 79401), CE 18:1 (cholesteryl oleate; Cat#: C9253), and cholesterol (Cat#: C3045) were obtained from Sigma Aldrich (St. Louis, MO). TG 54:0 (tristearin/tristearylglycerol; Cat#: 33–1800) and CE 18:0 (cholesteryl stearate; Cat#: 64–1800) were obtained from Larodan (Solna, SWE). Cholesterol [25,26–3H] (Cat#: ART-1987) was from American Radiolabeled Chemicals Inc (St. Louis, MO), [9,10–3H] Triolein was from PerkinElmer Life Sciences (Waltham, MA) and [1–14C] Palmitic acid (Cat#: MC121) was from Moravek Biochemicals (Brea, CA).

## Standards for lipid quantification by shotgun mass spectrometry

Synthetic lipid standards and ergosterol were purchased from Avanti Polar Lipids, Inc (Alabaster, AL), Sigma Aldrich (St. Louis, MO) and Toronto Research Chemicals (Toronto, CA). $^{13}$C uniformly labeled glucose was purchased from Euriso-top (Saint Aubin, FR) and yeast nitrogen base without amino acids from BD Difco (Le Pont de Claix, FR). All used solvents were of at least HPLC grade. Stocks of internal standards were stored in glass ampoules at −20℃ until used for the preparation of internal standard mix in 10:3 methyl-*tert*-butyl ether (MTBE)/methanol. A total of 700 µl internal standard mix contained: 356 pmol cholesterol $D_7$, 224 pmol zymosterol $D_5$, 215 pmol campesterol $D_6$, 207 pmol sitosterol $D_6$, 201 pmol lanosterol $D_6$, 418 pmol stigmasterol $D_6$, 233 pmol desmosterol $D_6$, 196 pmol $^{13}$C ergosterol, 443 pmol CE 16:0 $D_7$, 417 pmol TG 50:0 $D_5$, 116 pmol DG 34:0 $D_5$, 220 pmol PC 25:0, 77 pmol LPC 13:0, 107 pmol PS 25:0, 354 pmol PE 25:0, 85 pmol LPE 13:0, 96 pmol PI 25:0, 109 pmol PG 25:0, 145 pmol Cer 30:1, 123 pmol PA 25:0, 91 pmol LPA 13:0, 178 pmol CerPE 29:1, 38 pmol LPI 13:0, 54 pmol CL 56:4, 59 pmol LPS 13:0, 75 pmol LPG 13:0. $^{13}$C uniformly labeled ergosterol was produced in the prototrophic yeast strain W303 Y3358 according to *Knittelfelder et al., 2020*.

## Fly strains

The strains *Akh^A*, *bmm^1* and *FB-GAL4* have been described previously (*Gáliková et al., 2015*; *Grönke et al., 2007*; *Grönke et al., 2005*). *Hsl^1*, *UAS-Hsl* and *UAS-Hsl-EGFP* were generated in this study as described below. *Hsl^b24* was a kind gift of Xun Huang (*Bi et al., 2012*). *Act5c-GAL4* (Cat#: 4414) *Myo31DF-GAL4* (Cat#: 67088), *Desat1-GAL4* (Cat#: 65404), *elav-GAL4* (Cat#: 8760), *Cg-GAL4* (Cat#: 7011), and *nos-GAL4* (Cat#: 64277) strains were obtained from Bloomington *Drosophila* Stock Center (BDSC, Bloomington, IL). A *Hsl shRNA* transgenic strain and a control strain were obtained from the TRiP collection of BDSC (Cat#: 65148, Cat#: 36304). A *w^1118* strain (Cat#: 60000) from Vienna *Drosophila* Resource Center (VDRC, Vienna, AUT) was used as control and for backcrossing of transgenic flies. More details are described in *Table 1*.

**Table 1.** *Drosophila* strains used in the study.

| Name in text/figure | Full genotype | ID/Reference |
|---|---|---|
| Control or +/+ | $w^{1118}$; +; + | VDRC 60000 |
| $Hsl^1$ or -/- | $w^{1118}$; $Hsl^1$; + | This study |
| $Hsl^{b24}$ | $w^{1118}$; $Hsl^{b24}$; + | **Bi et al., 2012** |
| $Akh^A$ | $w^{1118}$; +; $Akh^A$ | **Gáliková et al., 2015** |
| $bmm^1$ | $w^{1118}$; +; $bmm^1$ | **Gáliková et al., 2017** |
| $Akh^A\ bmm^1$ | $w^{1118}$; +; $Akh^A\ bmm^1$ | This study |
| $Hsl^1\ bmm^1$ | $w^{1118}$; $Hsl^1$; $bmm^1$ | This study |
| $Hsl^1\ Akh^A$ | $w^{1118}$; $Hsl^1$; $Akh^A$ | This study |
| UAS-Hsl or +/+; UAS-Hsl | $w^{1118}$; +; $P\{w^{+mC}\ Hsl\ [Scer\backslash UAS]=UAS\text{-}Hsl\}/+$ | This study |
| Act5c> | $w^*$; $P\{w^{+mC}=Act5C\text{-}GAL4\}25FO1/+$; + | BDSC 4414 |
| Act5c>UAS-Hsl | $w^*$; $P\{w^{+mC}=Act5C\text{-}GAL4\}25FO1/+$; $P\{w^{+mC}\ Hsl\ [Scer\backslash UAS]=UAS\text{-}Hsl\}/+$ | This study, BDSC 4414 |
| $Hsl^1$ UAS-Hsl or -/- UAS-Hsl | $w^{1118}$; $Hsl^1$; $P\{w^{+mC}\ Hsl\ [Scer\backslash UAS]=UAS\text{-}Hsl\}/+$ | This study |
| $Hsl^1$ Act5c | $w^*$; $Hsl^1\ P\{w^{+mC}=Act5C\text{-}GAL4\}25FO1/Hsl^1$; + | This study, BDSC 4414 |
| $Hsl^1$ Act5c UAS-Hsl | $w^*$; $Hsl^1\ P\{w^{+mC}=Act5C\text{-}GAL4\}25FO1/Hsl^1$; $P\{w^{+mC}\ Hsl\ [Scer\backslash UAS]=UAS\text{-}Hsl\}/+$ | This study, BDSC 4414 |
| FB>+ or FB-GAL4 | $w^*$; $P\{w^{+mW.hs}=GawB\}FB/+$; + | **Grönke et al., 2003** |
| $Hsl^1$ FB | $w^*$; $Hsl^1\ P\{w^{+mW.hs}=GawB\}FB/Hsl^1$; + | This study |
| $Hsl^1$ FB UAS-Hsl | $w^*$; $Hsl^1\ P\{w^{+mW.hs}=GawB\}FB/Hsl^1$; $P\{w^{+mC}\ Hsl\ [Scer\backslash UAS]=UAS\text{-}Hsl\}/+$ | This study |
| $Hsl^1$ Myo31DF | $w^*$; $Hsl^1\ P\{w^{+mW.hs}=GawB\}Myo31DF^{NP0001}/Hsl^1$; + | This study, BDSC 67088 |
| $Hsl^1$ Myo31DF UAS-Hsl | $w^*$; $Hsl^1\ P\{w^{+mW.hs}=GawB\}Myo31DF^{NP0001}/Hsl^1$; $P\{w^{+mC}\ Hsl\ [Scer\backslash UAS]=UAS\text{-}Hsl\}/+$ | This study, BDSC 67088 |
| $Hsl^1$ Desat1 | $w^*$; $Hsl^1\ P\{w^{+mC}=Desat1\text{-}GAL4.E800\}2M/Hsl^1$; + | This study, BDSC 65404 |
| $Hsl^1$ Desat1 UAS-Hsl | $w^*$; $Hsl^1\ P\{w^{+mC}=Desat1\text{-}GAL4.E800\}2M/Hsl^1$; $P\{w^{+mC}\ Hsl\ [Scer\backslash UAS]=UAS\text{-}Hsl\}/+$ | This study, BDSC 65404 |
| $Hsl^1$ elav | $w^*$; $Hsl^1$; $P\{w^{+mC}=GAL4\text{-}elav.L\}3/+$ | This study, BDSC 8760 |
| $Hsl^1$ elav UAS-Hsl | $w^*$; $Hsl^1$; $P\{w^{+mC}=GAL4\text{-}elav.L\}3/\ P\{w^{+mC}\ Hsl\ [Scer\backslash UAS]=UAS\text{-}Hsl\}$ | This study, BDSC 8760 |
| -/-; nos-GAL4 | $w^*$; $Hsl^1$; $P\{w^{+mC}=GAL4::VP16\text{-}nos.UTR\}1C/+$ | This study, BDSC 64277 |
| -/-; nos-GAL4/UAS-Hsl | $w^*$; $Hsl^1$; $P\{w^{+mC}=GAL4::VP16\text{-}nos.UTR\}1C/\ P\{w^{+mC}\ Hsl\ [Scer\backslash UAS]=UAS\text{-}Hsl\}$ | This study, BDSC 64277 |
| Cg>+ | $w[1118]/y[1]\ v[1]$; $P\{w[+mC]=Cg\text{-}GAL4.A\}2/\ P\{y[+t7.7]=CaryP\}attP40$; + | BDSC 7011 and BDSC 36304 |
| Hsl shRNA/+ | $w[1118]/y[1]\ sc[*]\ v[1]\ sev[21]$; $P\{y[+t7.7]\ v[+t1.8]=TRiP.HMC05951\}attP40/+$; + | BDSC 65148 and VDRC 60000 |
| Cg>Hsl shRNA | $w[1118]/y[1]\ sc[*]\ v[1]\ sev[21]$; $P\{y[+t7.7]\ v[+t1.8]=TRiP.HMC05951\}attP40/\ P\{w[+mC]=Cg\text{-}GAL4.A\}2$; + | BDSC 7011 and BDSC 65148 |

## Generation of transgenic animals, genotyping, and mutant characterization

The *Hsl*[1] allele was generated by imprecise excision of a P-element inserted in the 5′ UTR of the *Hsl* gene (*GE15823*) using a conventional P-element mobilization procedure. Imprecise excision events were identified by PCR using 5′-AAAGATCTGAGCCGCAATAGGTGGAC-3′ and 5′- AAGGTACCC TGATGAAGCGGCTAGACTTG-3′ as flanking primers. This resulted in the identification of the *Hsl*[1] allele, which encompasses a deletion of 2743 bp (2R:20,257,711.20,260,454; FlyBase FB2020-04) that removes most of the *Hsl* open-reading frame including the start codon. Sequencing of the PCR products revealed the presence of 51 bp residual P element sequence in the *Hsl*[1] allele. The same PCR reaction was routinely used for genotyping of the *Hsl* locus (*Figure 2—figure supplement 1*). To this end, individual flies were crushed in 50 µl 10 mM Tris/HCl buffer pH 8.2 containing 1 mM EDTA, 25 mM NaCl, and 200 µg/ml proteinase K. Homogenates were incubated for 30 min at 37°C followed by 5 min at 95°C. Debris was removed by centrifugation for 1 min at 10,000 x *g* and 1 µl of the supernatant was subjected to PCR using the Phire Tissue Direct PCR Master Mix (Cat#: F170S; Thermo Scientific, Waltham, USA) using the following cycling conditions: 5 min 98°C; 35 cycles with 30 s 98°C, 30 s 55°C, 90 s 72°C; 10 min 72°C; hold at 4°C. The *Hsl*[1] allele was backcrossed for eight generations into the *w*[1118] background prior to analysis. Transgenic flies for the expression of *Hsl* under *UAS*-control were generated by P-element-mediated germline transformation. To construct plasmids for the expression of *Hsl* under the control of the *UAS* element a pFLC-1 plasmid encoding the *Hsl* cDNA (clone#: RE52776, *Drosophila* Genomics Resource Center, Bloomington, IL) was digested with NotI and KpnI and the fragment encoding *Hsl* was ligated into pUAST. A Hsl-EGFP fusion gene was engineered by PCR amplification of the *Hsl* cDNA with the primer pairs 5′- AAAGA TCTGAGCCGCAATAGGTGGAC-3′ and 5′- AAGGTACCCTGATGAAGCGGCTAGACTTG-3′ and subsequent ligation of the PCR product to the pEGFP-N2 vector by means of the KpnI and BglII restriction sites. The region encoding Hsl-EGFP was then cut with BglII and NotI and ligated to pUAST. Finally, pUAST plasmids were injected into embryos. Eye color was used to identify transgenic flies, which were backcrossed into the *w*[1118] background for six generations. Mice lacking the *Lipe/HSL* gene specifically in adipocytes (AHKO) were generated by crossbreeding *HSL*[flox/flox] mice with Adiponectin-Cre transgenic mice (The Jackson Laboratory, Bar Harbor ME; JAX stock number: 010803; genetic background: C57BL/6J) (*Eguchi et al., 2011*; *Haemmerle et al., 2002a*). Mice were backcrossed onto C57BL/6J for 10 generations. Mice homozygous for the *HSL*[flox/flox] allele were used as controls.

## Animal husbandry

Flies were maintained at 25°C, 60% humidity, and a 12 hr:12 hr dark/light cycle. If not stated otherwise, flies were reared on standard food containing 69.6 g/l corn flour (Haindl Mühle, Kalsdorf bei Graz, AUT), 8.7 g/l soy flour (Soja Austria, Vienna, AUT), 19.1 g/l molasses/beet syrup (Cat#: 01936, Grafschafter, Meckenheim, DE), 69.6 g/l malt (Cat#: 728985, CSM Austria, Vienna, AUT), 15.7 g/l yeast (Cat#: 03462, Gewürzmühle Brecht, Eggenstein, DE), 5.4 g/l agar-agar (Cat#: 00162, Gewürzmühle Brecht, Eggenstein, DE), 5.4 ml/l propionic acid (Cat#: P5561 Merck, Darmstadt, DE) and 1.3 g/l methyl 4-hydroxybenzoate (Cat#: W271004, Merck, Darmstadt, DE). Flies were starved on tap water containing 6 g/l agar-agar (Cat#: 2266.2, Carl Roth, Karlsruhe, DE). Lipid-depleted medium (LDM), which is naturally low in sterols and other lipids, contained 100 g/l yeast extract (Cat#: 212720; Bacto Thermo Fisher Scientific, Waltham, MA), 100 g/l sucrose, 6 g/l agar-agar (Cat#: 00162, Gewürzmühle Brecht, Eggenstein, DE), 5.4 ml/l propionic acid (Cat#: P5561, Sigma Aldrich, St. Louis, MO), 1.3 g/L methyl 4-hydroxybenzoate (Cat#: W271004, Sigma Aldrich, St. Louis, MO), and (optional) 0.01% cholesterol (w/v). For density seeding and embryo collections flies were reared on apple juice agar plates containing 20 g/l sucrose, 20% commercial apple juice, 17 g/l agar-agar (Cat#: 2266.2, Carl Roth, Karlsruhe, DE), and 1.3 g/l methyl 4-hydroxybenzoate. Plates were supplemented with fresh yeast paste to promote egg laying. If not otherwise indicated mated female flies were used for all assays. Mice were maintained under specific pathogen free conditions with ad libitum access to a chow diet (R/M-H Extrudate, Cat#: V1126-037, Ssniff Spezialdiäten GmbH, Soest, DE). Regular housing temperatures were maintained between 22–23°C with a 14 hr:10 hr light/dark cycle. Females were used for all analyses unless indicated otherwise. All animal experiments were approved by the Austrian Federal Ministry for Science, Research, and Economy (protocol number

BMWFW-66.007/0026/-WF/V/3b/2017) and the ethics committee of the University of Graz, and were conducted in compliance with the council of Europe Convention (ETS 123). Animals were allocated into experimental or control groups depending on the genotype. Individuals of the same genotype and age were randomly allocated to different experimetal conditions (e.g. food source).

## Expression of recombinant His$_6$-Hsl

For the expression of His$_6$-Hsl in tissue culture cells the open reading frame of the *Hsl* gene was amplified by PCR using the primer pair 5'-GCACTCGAGATTGACGCGGCTTCCG-3' and 5'- GCATC TAGACTATGAAGCGGCTAGACTTG-3' and ligated to the vector pcDNA4/Hismax C (Cat#: V86420, Thermo Fisher Scientific, Waltham, MA) using the restriction sites XhoI and XbaI. A plasmid for the expression of His$_6$-β-Galactosidase was provided by the supplier. A plasmid encoding for His$_6$-Hsl from *Mus musculus* (*Mm*Hsl) has been described previously (*Lass et al., 2006*). All plasmids were purified using the QIAGEN Plasmid Midi Kit (Cat#: 12143, QIAGEN, Hilden, DE). COS-7 cells (Cat#: CRL-1651; RRID: CVCL_0224; American Type Culture Collection, ATCC, Manassas, VA) were cultured in Dulbecco's modified Eagle's medium (Cat#: 41965039; Gibco Thermo Fisher Scientific, Waltham, MA) supplemented with 10% fetal bovine serum (FBS), 100 units/ml penicillin, and 100 µg/ml streptomycin at 37°C, 95% humidity, and 5% CO$_2$. *Mycoplasma* tests were routinely performed to ensure that cells are devoid of contamination. Cells were transfected with DNA complexed to Metafectene (Cat#: T020-5.0; Biontex GmbH, DE) according to the manufacturer's instructions and used for experiments 24 hr thereafter.

## Lipid hydrolase assays

Cell culture samples for lipid hydrolase assays were prepared by sonication of COS-7 cells in 0.25 M sucrose, 1 mM EDTA, 1 mM DTT containing 20 µg/ml leupeptin, 2 µg/ml antipain, and 1 µg/ml pepstatin (solution A) followed by centrifugation at 4°C and 1,000 × *g*. Fly abdominal samples were prepared by gross dissection of 10–15 abdomen per replicate in 250 µl solution A followed by sonication for 30 s with 15% output power using a Sonoplus sonicator (Bandelin electronic GmbH, Berlin, DE). These abdominal homogenates were first centrifuged at 4°C and 1000 x *g* for 10 min to remove debris. The resulting supernatants were then centrifuged at 4°C and 20,000 x *g* for 30 min and the soluble fraction was used for lipid hydrolase assays. Staged embryos were disrupted by sonication in solution A, centrifuged at 4°C and 1000 x *g* for 10 min and the resultant supernatants were used for lipid hydrolase assays. Murine tissue samples were homogenized in solution A using an UltraTurrax tissue homogenizer (IKA, Staufen, DE) and centrifuged at 20,000 x *g* and 4°C for 30 min to collect soluble infranatans. Protein concentrations were determined with the Bio-Rad Protein Assay kit according to the manufacturer's instructions (Cat#: 5000001; Bio-Rad Laboratories, Inc, Hercules, CA) using bovine serum albumin (BSA) as standard. Substrates for assaying TG and MG hydrolase activities were prepared as described using a Bis-tris propane buffer pH 7.5 (*Heier et al., 2016*; *Heier et al., 2015*). DG hydrolase substrate was prepared by emulsifying 0.3 mM 1,2-dioleoyl-*rac*-glycerol with 37 µM egg yolk PC and 11 µM soybean PI by sonication in BTP buffer pH 7.5 followed by the addition of essentially fatty acid free BSA to a final concentration of 2%. SE hydrolase substrate was prepared by emulsifying 0.45 mM cholesteryl oleate (containing 1 µCi/ml $^{14}$C-cholesteryl oleate as tracer) with 0.75 mM egg yolk phosphatidylcholine and 0.23 mM soybean phosphatidylinositol by sonication in BTP buffer pH 7.5 followed by the addition of essentially fatty acid free BSA to a final concentration of 5%. For TG and CE hydrolase assays, 100 µl samples were incubated with 100 µl substrate aliquots at 37°C for 60 min in a shaking water bath. Enzyme reactions were terminated by the addition of 3.3 ml methanol/chloroform/heptane (10:9:7, v/v/v) and 1 ml 0.1 M potassium carbonate/boric acid buffer pH 10.5. After centrifugation radioactivity in 1 ml of the upper phase was determined by liquid scintillation counting. MG and DG hydrolase activities were determined by incubating 10 µl protein samples with 50 µl substrate aliquots for 30–60 min at 37°C in a 96-well plate. The generation of free fatty acids was then determined with a colorimetric kit (NEFA–HR(2) reagent, Cat#: 999–34691, Cat#: 995–34791, Cat#: 991–34891, Cat#: 993–35191, and Cat#: 276–76491, Fujifilm Wako Diagnostics, Mountain View, CA). Mock substrates without MG or DG were used to determine background FA formation. All lipid hydrolase assays were repeated at least twice. All assays were performed with 3–8 biological replicates except for assays with recombinant proteins expressed in COS-7 cells, in which three technical replicates were used.

## Immunoblotting analysis

Protein extracts of transfected cells and murine tissue samples for immunoblotting analysis were generated as for the lipid hydrolase assays. Equal amounts of cell or tissue protein were separated by SDS-PAGE and electroblotted onto nitrocellulose membranes. Unspecific binding sites were blocked by incubating membranes with 10% milk powder in 150 mM NaCl, 50 mM Tris-HCl, 0.1% Tween-20, pH 7.6 (TBST) for 1 hr at room temperature under mild agitation. Proteins with N-terminal His$_6$-tags were detected by consecutive incubations with a monoclonal Anti-His antibody from mouse and an Anti-Mouse IgG-Horseradish Peroxidase antibody from sheep diluted 1:5,000 in TBST containing 5% milk powder. Murine HSL and GAPDH were detected using a polyclonal HSL antibody from rabbit (1:1,000) and a monoclonal Anti-GAPDH antibody from rabbit (1:10,000), respectively, as first antibodies, and a polyclonal Anti-Rabbit IgG (H+L)-Horseradish Peroxidase antibody from goat (1:5,000) as secondary antibody. Chemiluminescence was detected using the Pierce Supersignal West Pico PLUS chemiluminescent substrate (Cat#: 34577, Thermo Fisher Scientific, Waltham, MA) and the Bio-Rad ChemiDoc imaging system (Bio-Rad Laboratories, Inc, Hercules, CA).

## Physiological assays

For metabolite assays, microcalorimetry, and fecundity assays 1st instar larvae were collected on apple juice agar plates and grown at non-crowding densities in standard food (~50 animals per 7 ml food in a 28 ml vial or ~150 animals per 20 ml food in a 68 ml vial). After eclosion, flies were maintained in 68 ml vials at densities of 50 males plus 50 females for the indicated time with regular transfer to fresh food. Flies were frozen in liquid nitrogen and either processed immediately or stored at −80°C. The determination of glycogen and glucose was performed according to Tennessen et al. using the Glucose (GO) Assay kit (Cat#: GAGO20-1KT, Sigma Aldrich, St. Louis, MO) and Amyloglucosidase (Cat#: A1602-25MG, Sigma Aldrich, St. Louis, MO) (*Tennessen et al., 2014*). Glycerol was determined from the same homogenates using the Free Glycerol Reagent (Cat#: F6428, Sigma Aldrich, St. Louis, MO). TG equivalents were determined according to Hildebrandt et al. using the Triglycerides reagent (Cat#: 981786, Thermo Fisher Scientific, Waltham, MA) (*Hildebrandt et al., 2011*). Free FA were determined from the same homogenates as TG using the NEFA-HR(2) reagent (Cat#: 999–34691, Cat#: 995–34791, Cat#: 991–34891, Cat#: 993–35191, and Cat#: 276–76491, Fujifilm Wako Diagnostics, Mountain View, CA). For microcalorimetry measurements five adult flies or 100 embryos were transferred to 2 ml glass ampoules supplemented with 500 µl complex medium or agar, respectively, and heat dissipation was measured in a TAM IV microcalorimeter (TA instruments, New Castle, DE). Flies were pre-starved for 8 hr before measurements of the starved status. For determination of hatching rates on apple juice agar plates with yeast paste batches of ~100 embryos were collected 2 hr AEL and scored for hatching events 24–48 hr later. For fecundity assays, single female flies were collected within 1d after eclosion and maintained with two males in 28 ml vials containing 2 ml food. Flies were flipped daily. Eggs were counted directly after each transfer and hatching rates were determined 48 hr later. To monitor starvation sensitivity, female flies were aged on standard food for 7d and then transferred to 68 ml vials containing 10 ml tap water with 0.6 g/l agar-agar (Cat#: 2266.2, Carl Roth, Karlsruhe, DE). 4 replicates with 10 flies were set up and dead flies were scored every 2–12 hr. All metabolite and microcalorimetry assays were performed at least twice with 4–12 biological replicates per experiment. Starvation experiments were performed at least twice with a total of 39–40 flies in four replicates. Hatching rates were determined twice with ~500 individual embryos in five replicates. Fecundity assays were typically performed with 10–17 biological replicates per genotype and data of two independent experiment was pooled resulting in a total of 25–34 biological replicates. Individual flies or cohorts were excluded from analysis upon the following criteria: (1) complete sterility, (2) early death unrelated to the applied stress (e.g. squeezing or sticking to food), (3) early escape from experimental vial.

## Tissue staining, immunohistochemistry, and confocal laser scanning microscopy

Abdominal fat body tissue associated with the spermatheca of ad libitum fed females was dissected in ice-cold phosphate-buffered saline (PBS) and incubated for 30 min in PBS containing 5 µg/ml Hoechst 33342 (Cat#: 14533, Sigma Aldrich, St. Louis, MO), 5 µg/ml BODIPY 493/503 (Cat#: B2103, Thermo Fisher Scientific Invitrogen, Carlsbad, CA) and 1 µg/ml CellMask Deep Red (Cat#: C10046,

Thermo Fisher Scientific Life Technologies, Carlsbad, CA). Fat body tissue expressing Hsl-EGFP was incubated for 30 min in LipidTOX Deep Red (Cat#: H3477, Thermo Fisher Scientific Life Technologies, Carlsbad, CA) diluted 1:1000 in PBS. Samples were then mounted in PBS and imaged immediately by confocal fluorescence microscopy. Ovaries were dissected in PBS and fixed for 30 min in PBS containing 4% paraformaldehyde. Ovaries were then incubated for 30 min in PBS containing 5 µg/ml Hoechst 33342, 5 µg/ml BODIPY 493/503 and 1 µg/ml CellMask Deep Red. After transfering to glass slides, egg chambers were separated with fine forceps and mounted in PBS containing 30% glycerol. Ring glands of white prepupae were processed and stained as ovaries and were mounted in PBS. Samples were imaged on a Leica SP8 confocal laser scanning microscope using a HC PL APO 20x multi-immersion objective with a 0.75 NA. Embryos were collected on apple juice agar plates supplemented with yeast and aged at 25°C for the indicated time. Embryos were dechorionated with 50% bleach (DanKlorix Hygienereiniger mit Chlor), washed extensively with water and fixed for 20 min in a mixture of 500 µl heptane and 500 µl PBS with 4% paraformaldehyde. After removal of the aqueous phase, 1 ml methanol was added and samples were vortexed for 1 min. Embryos were washed twice with methanol and three times for 5 min with 0.5 ml 10 mM Tris pH 7, 55 mM NaCl, 7 mM $MgCl_2$, 5 mM $CaCl_2$, 20 mM glucose, 50 mM sucrose, 0.1% BSA, and 0.1% Tween-20 (BBT). Unspecific binding sites were blocked by incubating samples for 1 hr with 0.5 ml BBT containing 2% horse serum. Primary antibodies were diluted 1:100 in 100 µl BBT and incubated with the samples over night at 4°C. Primary antibody binding was detected using the Vectastain Elite ABC-HRP system according to the manufacturer's instructions (Cat#: PK-6200, Vector Laboratories, Peterborough, UK) with 3,3'-diaminobenzidine as peroxidase substrate.

## Extraction of mRNA and qPCR

RNA was isolated from cohorts of 5 adult female flies using the RNeasy Mini Kit (Cat#: 74104, QIAGEN, Hilden, DE). Murine adipose tissue (~100 mg) was homogenized in 1 ml of TRIzol reagent (Cat#: 15596026, Thermo Fisher Invitrogen, Carlsbad, CA) using an Ultra-Turrax homogenizer (IKA, Staufen, DE) and incubated at room temperature for 5 min. Phase separation was induced by addition of 100 µl 1-bromo-3-chloropropane and centrifugation at 12,000 x $g$ and 4°C for 15 min. Clear supernatant was transferred and total RNA was precipitated by addition of 500 µl isopropyl alcohol and centrifugation at 12,000 x $g$ and 4°C for 15 min. cDNA was synthesized using the SuperScript III First-Strand Synthesis Supermix (Cat#: 18080051, Thermo Fischer Invitrogen, Carlsbad, CA) for fly samples and QuantiTect Reverse Transcription Kit (Cat#: 205313, QIAGEN, Hilden, DE) for murine samples. qPCR was performed on the StepOnePlus Real-Time PCR System (Applied Biosystems, Waltham, MA) using iTaq Universal SYBR Green Supermix (Cat#: 1725120, Bio-Rad, Hercules, CA). The thermal program for the qPCR included stage 1: 95°C, 10 min and stage 2: 95°C, 0.5 min and 60°C, 1 min for a total of 40 cycles. Non-template control (RNA) and non-reaction control ($H_2O$) were routinely performed. qPCR experiments were performed twice with 4–5 biological replicates. Relative mRNA expression levels were determined by means of the ΔΔCt method using *rp49* (FlyBase name *RpL32*) and *36B4* as normalization control for fly and murine samples, respectively. A comprehensive list of primers used can be found in *Table 2*.

## Lipid extraction and thin-layer chromatography (TLC)

For the determination of total SE and TG content, lipid extraction followed essentially the protocol of Matyash et al. with minor modifications (*Matyash et al., 2008*). Batches of 5 flies, five adult segments, five adult tissues, 100 embryos, or pieces of 10–20 mg murine adipose tissue were collected in 2 ml safe-seal micro tubes (Cat#: 72.695.500, Sarstedt, Nürmbrecht, DE) and were disrupted in a vibration mill (Retsch MM400, Retsch, Haan, DE) with a metal bead (5 mm diameter; Cat#: 504942, Askubal Korntal-Münchingen, DE) in 1 ml MTBE/methanol (10:3, v/v) for 60 s followed by incubation at 600 rpm and 25°C for 30 min in a ThermoMixer (Eppendorf, Hamburg, DE). After the addition of 200 µl distilled $H_2O$ samples were vortexed and phase separation was induced by centrifugation at 16,000 x $g$ for 5 min. Upper phases were collected and lower phases were re-extracted with 300 µl artificial upper phase. The combined upper phases were evaporated in a stream of nitrogen, dissolved in chloroform/methanol (2:1, v/v) and aliquots of the extracts were applied onto silica gel 60-coated TLC plates (Cat#: 1.05554.0001, Merck, Darmstadt, DE), which were developed in petrolether/diethylether/acetic acid (25:25:1, v/v/v) until the solvent front reached ~1/3 of the plate. The

**Table 2.** List of primers used for RT-qPCR.

| Gene symbol | Primer sequences | ID/Reference |
|---|---|---|
| rp49/RpL32 | fw: 5'-CTTCATCCGCCACCAGTC-3' rv: 5'-CGACGCACTCTGTTGTCG-3' | |
| plin1 | fw: 5'-GCGCGAATTCTGGCGCCCCTAGATG-3' rv: 5'-CACAGAAGTAAGGTTCCTTCACAAAGATCC-3' | |
| plin2 | fw: 5'-TCAAATTGCCCGTGGTAAA-3' rv: 5'-CCCATTCGAAGACACGATTT-3' | |
| Akh | - | QIAGEN QuantiTect QT00957859 |
| bmm | - | QIAGEN QuantiTect QT00964460 |
| Hr96 | fw: 5'-CCAGCGAGGCTCTTTATGAT-3' rv: 5'-GGTTGTGGCGAGTGTCGT-3' | |
| Npc1a | fw: 5'-GTCGAGGAACTTTGCAGGGA-3' rv: 5'-TCATCGAAACAGGACTGCGT-3' | |
| Npc1b | fw: 5'-CGGATTTTGTTCCAGCAACT-3' rv: 5'-CCATTCTCAGTAAATCCTCGTTC-3' | |
| Npc2a | fw: 5'-ACAGTCGTCCACGGCAAG-3' fw: 5'-ACACAGGCATCGGGATTG-3' | |
| Npc2b | fw: 5'-GGAGATCCACTGGGGATTG-3' rv: 5'-CCTTGATTTTGGCGGGTAT-3' | |
| CG8112 | fw: 5'-CACAAACTGAAACCGCACAG-3' rv: 5'-CGACACGAAACAGAAGACCA-3' | |
| magro | fw: 5'-ACACCGAACTGATTCCGAAC-3' rv: 5'-ATCCACCATTGGCAAACATT-3' | |
| Hsl | fw: 5'-CTGGAGGCGACCTATGGAAC-3' rv: 5'-GCTCGTCAAAATCGTACTCGTG-3' | |
| PCNA | fw: 5'-GCGACCGCAATCTCTCCAT-3' rv: 5'-CGCCTTCATCGTCACATTGT-3' | |
| Ate1 | fw: 5'-GCATACTTCGCCGCATAAATCG-3' rv: 5'-CTATGGCGTAATCGGCATCGG-3' | |
| MmAbca1 | fw: 5'-GATGTGGAATCGTCCCTCAGTTC-3' rv: 5'-ACTGCTCTGAGAAACACTGTCCTCC-3' | |
| MmSoat1 | fw: 5'-GAAACCGGCTGTCAAAATCTGGR-3' rv: 5'-TGTGACCATTTCTGTATGTGTCC-3' | |
| MmHmgcs1 | fw: 5'-GACAAGAAGCCTGCTGCCATA-3' rv: 5'-CGGCTTCACAAACCACAGTCT-3' | |
| MmHmgcr | fw: 5'-TGCACGGATCGTGAAGACA-3' rv: 5'-GTCTCTCCATCAGTTTCTGAACCA-3' | |
| MmSrebp2 | fw: 5'-GCGCCAGGAGAACATGGT-3' rv: 5'-CGATGCCCTTCAGGAGCTT-3' | |
| MmStar | fw: 5'-TTGGGCATACTCAACAACCA-3' rv: 5'-GAAACACCTTGCCCACATCT-3' | |
| Mm36B4 | fw: 5'-GCTTCATTGTGGGAGCAGACA-3' rv: 5'-CATGGTGTTCTTGCCCATCAG-3' | |

plates were briefly dried and further developed in petrolether/diethylether (49:1, v/v) until the solvent front reached the top of the plate. The plates were dried thoroughly and immersed in a derivatization solution containing 50 g/l copper sulfate, 10% phosphoric acid, and 25% ethanol. Carbonization of lipids was achieved by incubating plates for 25 min at 120°C. SE content was determined by scanning densitometry at 400 nm using a Camag TLC scanner 3 (Camag, Muttenz, CH) and 3+1 mixtures of cholesteryl oleate and cholesteryl stearate or trioleoylglycerol and tristearylglycerol, respectively, as references. Data were normalized to animal number (adult flies, fly embryos) or mg tissue weight (murine adipose tissue). Lipid analysis by TLC was performed at least twice with 3–8 biological replicates per experiment.

## Radiolabeling and tracer studies

For radiolabeling experiments, standard food was melted, mixed with $^3$H-cholesterol (0.05% final concentration and 50 µCi/ml final specific activity) or $^{14}$C-palmitic acid (200 µM final concentration and 10 µCi/ml final specific activity), and transferred à 500 µl into 24-well plates. 1st instar larvae were collected on apple juice agar plates supplemented with yeast paste and 20 larvae per genotype were transferred into each well containing radiolabeled food. Adult flies were sampled either within 1 day after eclosion or reared on unlabeled food for the indicated time. For lipid extraction, single flies were crushed with a pipet tip in 500 µl MTBE/methanol (3:1, v/v) and incubated for 30 min at 25°C and 600 rpm. After the addition of 100 µl $H_2O$, samples were vortexed and phase separation was induced by centrifugation at 16,000 x *g* for 5 min. Upper phases were collected and lower phases were re-extracted with 150 µl artificial upper phase. The combined upper phases were evaporated in a stream of nitrogen and the extracts were applied onto Silica G TLC plates, which were developed as described above. For the detection of sterols and SE, samples were spiked with

cholesterol and cholesteryl oleate. Lipid bands were detected by exposure to iodine vapor and the associated radioactivity was determined by liquid scintillation counting. For the measurement of cholesterol transfer to embryos, individual females were mated with two males of the same genotype in vials containing 2 ml LDM. Flies were flipped daily to fresh food and their egg cohorts were harvested in 100 µl distilled water. Embryos were disrupted by sonication and lipids were extracted by incubating the embryo homogenates with 500 µl MTBE/methanol (3:1, v/v) at 25°C and 600 rpm for 30 min. After centrifugation at 16,000 x *g* for 5 min the upper phases were collected and the lower phases were re-extracted with 150 µl artificial upper phase. Radioactivity in the combined upper phases was determined by liquid scintillation counting. Radiolabling experiments were performed at least twice with 4–5 biological replicates of adult flies and 9–10 replicates of embryos per assay. Individual flies or cohorts were excluded from analysis upon the following criteria: (1) complete sterility, (2) early death unrelated to the applied stress (e.g. squeezing or sticking to food), (3) early escape from experimental vial.

## LC-MS analysis of TG composition

Four batches of 5 flies each were mixed with 700 µl MTBE/methanol (10/3, v/v) in 2 ml safe-seal micro tubes (Cat#: 72.695.500, Sarstedt, Nürmbrecht, DE) and disrupted with a metal bead (5 mm diameter; Cat#: 504942, Askubal Korntal-Münchingen, DE) in a Retsch MM 400 mixer mill (3 min, 30 Hz, 4°C) and lipids were extracted by shaking for 20 min at 1400 rpm and 4°C in a ThermoMixer (Eppendorf, Hamburg, DE). After the addition of 200 µl $H_2O$, samples were again incubated 20 min at 1400 rpm and 4°C. Phase separation was induced by centrifugation for 10 min at 16,000 x *g* and 4°C. The upper organic phase was collected and dried under a stream of nitrogen. Lipids were washed by dissolving in 500 µl chloroform/methanol (2/1, v/v) and again dried under a stream of nitrogen and stored at −20°C or prepared immediately for LC-MS analysis. For MS analysis, lipid extracts were dissolved in 0.5 ml chloroform/methanol (2/1, v/v), 50 µl of the sample were diluted with 100 µl isopropanol and 10 µl were injected for analysis using UPLC-QTOF-MS as described (*Knittelfelder et al., 2014*). Data analysis was performed using the Lipid Data Analyzer (LDA) software (*Hartler et al., 2011*). The abundance of each TG species was normalized to the intensity of the internal standard TG 51:0 (Larodan, Solna, SWE) and to animal number.

## Lipid extraction and quantification by shotgun mass spectrometry

A total of 100 embryos or five female flies were homogenized with 1 mm zirconia beads in a cooled tissuelyzer for 2 × 5 min at 30 Hz in 200 µl isopropanol. The embryo homogenate was evaporated in a vacuum desiccator to complete dryness and subjected to lipid extraction. In case of the fly homogenate, an aliquot corresponding to one fly was transferred to a new sample tube and vacuum dried. Lipid extraction was performed according to *Sales et al., 2017*. In brief, 700 µl internal standard mix in MTBE/methanol (10:3, v/v) was added to each sample and vortexed for 1 hr at 4°C. After the addition of 140 µl $H_2O$, samples were vortexed for another 15 min. Phase separation was induced by centrifugation at 16,000 x *g* for 15 min. The organic phase was transferred to a glass vial and evaporated. Samples were reconstituted in 300 µl chloroform/methanol (2:1, v/v). To a new vial, 100 µl were transferred and used for lipidome analysis. To quantify sterols, 150 µl of lipid extract were evaporated and acetylated with 300 µl chloroform/acetyl chloride (2:1, v/v) for 1 hr at room temperature (modified from *Liebisch et al., 2006*). After evaporation, sterol samples were reconstituted in 150 µl isopropanol/methanol/chloroform (4:2:1, v/v/v) and 7.5 mM ammonium formate (spray solution). For sterol measurements, samples were 1:5 diluted with spray solution. For lipidome measurements, samples were 1:10 diluted with spray solution. Mass spectrometric analysis was performed on a Q Exactive instrument (Thermo Fischer Scientific, Waltham, MA) equipped with a robotic nanoflow ion source TriVersa NanoMate (Advion BioSciences, Ithaca, NY) using nanoelectrospray chips with a diameter of 4.1 µm. The ion source was controlled by the Chipsoft 8.3.1 software (Advion BioSciences, Ithaca, NY). Ionization voltage was + 0.96 kV in positive and − 0.96 kV in negative mode; backpressure was set at 1.25 psi in both modes. Samples were analyzed by polarity switching (*Schuhmann et al., 2012*). The temperature of the ion transfer capillary was 200°C; S-lens RF level was set to 50%. All samples were analyzed for 10 min. FT MS spectra were acquired within the range of m/z 400–1000 from 0 min to 0.2 min in positive and within the range of m/z 350–1000 from 5.2 min to 5.4 min in negative mode at the mass resolution of R m/z 200 = 140,000; automated gain

control (AGC) of $3 \times 10^6$ and with the maximal injection time of 3000 ms. *t*-SIM in positive (0.2–5 min) and negative (5.4–10 min) mode was acquired with R m/z 200 = 140,000; automated gain control of $5 \times 10^4$; maximum injection time of 650 ms; isolation window of 20 Th and scan range of m/z 400–1000 in positive and m/z 350–1000 in negative mode, respectively. The inclusion list of masses targeted in *t*-SIM analyses started at m/z 355 in negative and m/z 405 in positive ion mode and other masses were computed by adding 10 Th increment (i.e. m/z 355, 365, 375, . . .) up to m/z 1005. Acetylated sterols were quantified by parallel reaction monitoring (PRM) FT MS/MS in an additional measurement. FT MS spectra within the range of m/z 350–1000 were acquired from 0 min to 0.2 min and *t*-SIM ranging from m/z 350 to 500 were acquired from 0.2 min to 4 min with the same settings as described above. PRM spectra were acquired from 4 min to 10 min. For PRM micro scans were set to 1, isolation window to 0.8 Da, normalized collision energy to 12.5%, AGC to $5 \times 10^4$ and maximum injection time to 3000 ms. All spectra were pre-processed using repetition rate filtering software PeakStrainer and stitched together by an in-house developed script (*Schuhmann et al., 2017b*; *Schuhmann et al., 2017a*). Lipids were identified by LipidXplorer software (*Herzog et al., 2012*). Molecular Fragmentation Query Language (MFQL) queries were compiled for acetylated sterols, PA, LPA, PC, PC O-, LPC, LPC O-, PE, PE O-, LPE, PI, LPI, PS, LPS, PG, LPG, CL, CerPE, Cer, TG, DG lipid classes. The identification relied on accurately determined intact lipid masses (mass accuracy better than 5 ppm). Lipids were quantified by comparing the isotopically corrected abundances of their molecular ions with the abundances of internal standards of the same lipid class. For acetylated sterols, the specific fragment (loss of acetyl group) was used for quantification. Shotgun lipidomics experiments were performed twice with 3–4 biological replicates and both datasets were pooled for analysis resulting in a total number of 7–8 biological replicates. Data were normalized to animal number.

## Statistical analysis

Group differences were analyzed by Student's unpaired *t*-tests and one-way ANOVAs using the Bonferroni correction for multiple comparisons. Differences were considered statistically significant at $p < 0.05$. Lipidomic data were analyzed by multiple *t*-tests using the Benjamini-Hochberg method to control for a false discovery rate of 0.01. Survival was analyzed by log-rank tests. All statistical analyses were performed with the GraphPad Prism 8.0 software.

## Acknowledgements

The authors acknowledge Lisa Pichler for technical support. Transgenic stocks were obtained from the Vienna *Drosophila* Resource Center (VDRC, http://www.vdrc.at) and the Bloomington *Drosophila* Stock Center (NIH P40OD018537). The authors are indebted to the support of the University of Graz. SG and RPK acknowledge the support of the department Molecular Developmental Biology at their former institute, the Max Planck Institute for Biophysical Chemistry. G.S. is supported by the Austrian Science Fund FWF stand-alone project P28882-B21. IP is supported by the Austrian Science Fund FWF Lise Meitner Fellowship M 2706-B34. Work in AS laboratory was supported by FOR 2682 and TRR83 (TP17) grants from Deutsche Forschungsgemeinschaft (DFG).

## Additional information

### Funding

| Funder | Grant reference number | Author |
| --- | --- | --- |
| Austrian Science Fund | P28882-B21 | Gabriele Schoiswohl |
| Deutsche Forschungsgemeinschaft | FOR 2682 | Andrej Shevchenko |
| Deutsche Forschungsgemeinschaft | TRR83 (TP17) | Andrej Shevchenko |
| Austrian Science Fund | M 2706-B34 | Ingrid Pörnbacher |

The funders had no role in study design, data collection and interpretation, or the decision to submit the work for publication.

## Author contributions
Christoph Heier, Conceptualization, Data curation, Formal analysis, Supervision, Validation, Investigation, Visualization, Methodology, Writing - original draft, Project administration, Writing - review and editing; Oskar Knittelfelder, Data curation, Formal analysis, Investigation, Methodology, Writing - review and editing; Harald F Hofbauer, Investigation, Methodology, Writing - review and editing; Wolfgang Mende, Ingrid Pörnbacher, Laura Schiller, Hao Xie, Investigation; Gabriele Schoiswohl, Investigation, Writing - review and editing; Sebastian Grönke, Resources; Andrej Shevchenko, Resources, Methodology, Writing - review and editing; Ronald P Kühnlein, Resources, Supervision, Writing - review and editing

## Author ORCIDs
Christoph Heier ⦿ https://orcid.org/0000-0001-6858-408X
Oskar Knittelfelder ⦿ http://orcid.org/0000-0002-1565-7238
Harald F Hofbauer ⦿ http://orcid.org/0000-0003-2617-5901
Sebastian Grönke ⦿ https://orcid.org/0000-0002-1539-5346
Andrej Shevchenko ⦿ http://orcid.org/0000-0002-5079-1109

## Ethics
Animal experimentation: All animal protocols were approved by the Austrian Federal Ministry for Science, Research, and Economy (protocol number BMWFW-66.007/0026/-WF/V/3b/2017) and the ethics committee of the University of Graz, and were conducted in compliance with the council of Europe Convention (ETS 123).

## Decision letter and Author response
Decision letter https://doi.org/10.7554/eLife.63252.sa1
Author response https://doi.org/10.7554/eLife.63252.sa2

# Additional files

## Supplementary files
• Supplementary file 1. Composition of the lipidome.
• Transparent reporting form

## Data availability
All data generated or analysed during this study are included in the manuscript and supporting files.

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
