## [Decision Letter]

[Editors' note: this paper was reviewed by Review Commons.]

**Acceptance summary:**

Your results obtained in *Drosophila* provide new insights on the role of HSL and metabolism. More broadly, your paper will be of interest to fields of metabolism and reproduction and to the readership of *eLife*.

**Decision letter after peer review:**

Thank you for submitting your work entitled "Hormone-sensitive lipase is a major adipocyte steryl ester hydrolase and couples intergenerational sterol metabolism to reproductive success" for further consideration by *eLife*. Your revised article has been evaluated by two Reviewing Editors and a Senior Editor.

The manuscript has been improved but there are some remaining issues that need to be addressed before acceptance, as outlined below:

Specifically, the authors need to emphasize in the discussion that their findings are novel in the fly. In particular, the authors need to temper their statements regarding novelty in the mouse studies. It should also be noted that it is already well appreciated that HSL is a cholesterol ester hydrolase. This is well documented and known for many years (e.g., https://doi.org/10.1186/1743-7075-3-12). The manuscript makes it seem the authors discovered this fact and overstate the advance in this regard. For example, Figure 4E,F is not surprising at all based on this previous knowledge, and they didn't measure DAG which is also greatly increased just like CE is as shown. So the text should be more grounded in what is already known, which is that HSL is already a well know sterol esterase.

In addition, the authors should re-word "Generation of transgenic animals…" etc. to "Generation of transgenic flies…", as they otherwise refer to "fly strains" and the reader might be confused that this is vertebrate (animal) study. Also, statements such as "In summary, our study identifies an ancestral function of Hsl-related enzymes in sterol handling and illustrates how this enzyme orchestrates intergenerational sterol transfer to optimize reproductive success. " should clearly indicate that the work was done in flies.

---

## [Author Response]

Reviewer #1 (Evidence, reproducibility and clarity (Required)):In this study, Heier and colleagues identify Hormone-sensitive lipase (Hsl) as a major regulator of steryl ester (SE) hydrolysis in *Drosophila* and in mouse adipocytes. in vitro, Hsl has the ability to hydrolise many different classes of lipids, but in vivo Hsl mutants specifically have elevated SE levels. Quite impressively, knockout of HSL in mouse fat tissue causes SE levels to increase 15-fold in adipocytes. Physiologically, in *Drosophila*, Hsl activity in the fat body appears to be required to mobilize cholesterol for loading into eggs during vitellogenesis, for maximal viability of the eggs and ensuing embryonic development.Overall this study is very solid and the data are beautiful. The interpretations of the results are in line with the data. The key conclusions are convincing, so I have no major suggestions relating to the main claims of the paper.

We are grateful for this very positive perception of our study.

I have only minor suggestions:1) Title: it seems to me that one of the key findings of this manuscript is that Hsl is a major steryl ester hydrolase conserved from flies to mice, and this is not currently reflected in the title. This seems more important to me than the intergenerational aspect currently mentioned in the title (after all, there are probably many proteins, transcripts and metabolites that need to be loaded into the embryo during vitellogenesis for optimal embryonic development…). So I suggest the authors consider an alternative title such as "Hormone sensitive lipase is the main steryl ester hydrolase in adipocytes in flies and mammals"? This might also make the manuscript more obviously relevant for a broader audience when it shows up on pubmed.

We agree with the reviewer that functional conservation between mouse and fly Hsl is a major theme in our manuscript. However, due to character restrictions we are unable to include this important aspect in the title. As pointed out by Reviewer 2, the major novelty of our study is the impact of (adipose) Hsl on reproduction and thus we believe that the title should reflect this aspect.

2) Figure 1A – it would be good to have a non-transfected negative control to show that the band in the 2nd lane (HIS6-Hsl) is not a background band. (Admittedly, this is unlikely to be the case given that HIS6-Hsl expression is increasing lipase activity, as seen in Figure 1B, however without a negative control Figure 1A is not very useful.)

We included transfections with murine Hsl in the experiments depicted in Figure1A-B. The new data show that the discussed band is not a background band.

3) Figure 1D- it would be helpful for the color image to indicate instead of "merge" written in alternating colors that GFP is green and lipid droplets are red.

We agree with the reviewer and improved Figure 1D as proposed.

4) Figure 2A – it would be good to show the molecular characterization of the knockout allele (e.g. genomic PCR across the junction, or whatever was done to molecularly characterize the deletion).

We thank the reviewer for this suggestion. Now we included the result of a representative PCR genotyping experiment in the new Figure 2—figure supplement 1 and added the detailed sequence information regarding the molecular characterization of the allele to the Materials and methods section. Moreover, we revised Figure 2A to provide a more precise representation of the

Hsl1 allele.

5) Q-RT-PCR would be good on the Hsl homozygous knockout flies to show that Ate1 and PCNA expression are not altered (given that they are nearby, and the deleted region could harbor regulatory elements).

Both, Ate1 and PCNA are essential genes arguing against a major functional impairment in the homozygous viable Hsl1 mutant flies. To confirm this, we measured mRNA levels of Ate1 and PCNA in Hsl1 and control animals by qRT-PCR and included the data in Figure 2—figure supplement 1

6) Why are SE levels in Hsl1 mutants at eclosion not elevated compared to controls? Does this suggest that during larval development very little SE hydrolysis is going on and the animal mainly stores SEs?

Under our experimental conditions net SE formation occurs during metamorphosis likely as a strategy to sequester excess dietary and tissue sterol during histolysis. Hsl activity may be limited under such conditions to prevent a futile cycle of SE formation and hydrolysis. However, since Hsl function during metamorphosis was not experimentally addressed in this study we do not to include this speculation in the manuscript.

7) Figure 7A – it would be useful to also show L1 larvae per egg laid (ie the ratio of 1st instar to eggs) as a readout for possible phenotypes during embryonic development. This “hatching rate” is mentioned in the text, but I believe it's not currently shown?

We present hatching rates of control and Hsl1 mutant animals in the new Figure 7—figure supplement 1 and refer to this result in the main text. These results show that hatching rates are 14% lower in Hsl1 vs control animals after 4d on LDM.

Reviewer #1 (Significance (Required)):Sterols such as cholesterol are present in animals both in the free form, and esterified to fatty acids. The balance of steryl ester synthesis and hydrolysis is important for proper homeostasis. This manuscript makes the important discovery that Hsl is a steryl ester hydrolase in both flies and mice. The phenotype in Figure 4F is very impressive – it shows that HSL is actually a major regulator of SE breakdown in mammalian adipocytes. Hence even though other SE hydrolases have been identified in mammals, Hsl appears to be an important one in adipose tissue.Overall, I think this is an important discovery for the metabolism field.Reviewer background: *Drosophila* metabolism and developmentReviewer #2 (Evidence, reproducibility and clarity (Required)):This study explores the functions of the single Hormone-sensitive lipase (Hsl) in *Drosophila* melanogaster. Hsl-like proteins are present from bacteria to humans and are generally believed to degrade a wide range of neutral lipid including triglyceride (TG). The authors' data somewhat challenges this view by revealing physiological specificity for steryl ester (SE) degradation in vivo in Drosophila, and reveals a specific function for Hsl in mediating sterol transfer from mothers to progeny.In brief, the authors specifically show that:1) Hsl hydrolyses a broad range of lipid esters in vitro, and it localizes to lipid droplets when ectopically expressed in *Drosophila* fat body (analogous to liver/adipose tissue)2) Hsl mutants have normal TG levels and energy metabolism, but accumulate SE because of defects in SE catabolism3)The SE accumulation of Hsl mutants is specifically rescued from the fat body, and mouse adipose tissue from adipose-specific Hsl KO mice also accumulates SE4) Embryonic SE catabolism depends on the maternal Hsl genotype, and re-expression of Hsl form the germline of Hsl mutant mothers rescues the SE catabolism phenotype of their progeny5) Hsl mutant females transfer less labelled cholesterol to the developing egg and produce fewer viable eggs in lipid-depleted medium. Egg viability is partially rescued by supplementing this maternal diet with cholesterolCollectively, the data are detailed and comprehensive, and the authors convincingly show that Hsl acts as specific regulator of sterols but not of other lipids. The manuscript could be published in its current form with some minor amendments (listed below) that will improve data presentation and/or the manuscript's readability. But I have listed a couple of experimental suggestions that will increase the general interest of the manuscript and will strengthen its more novel findings.

We thank this reviewer for this very positive appreciation of our study and for the selective proposals for improvement, which we will largely follow.

Experimental suggestions (in order of importance)1) The authors hypothesise, but do not demonstrate, that fat body SE catabolism may be "upstream" of the embryogenesis phenotypes. This could easily be tested by, for example, downregulating Hsl in fat body and assessing fecundity in a lipid depleted diet, and/or by testing whether the fecundity phenotype of Hsl mutants can be rescued. It would be important to do so because this intergenerational phenotype is novel and arguably one of the more interesting findings reported in the manuscript. Related to this, to shed further light on the Hsl mutant rescue from nos-Gal4, the authors could have established what exactly is transferred to the embryo (i.e. Hsl itself and/or only lipid species) through, for example, germline mutant clones.

According to the reviewer’s suggestions we tested fecundity of (1) animals with fat bodyspecific downregulation of Hsl expression and (2) Hsl1 mutant animals with fat-body specific re-expression of transgenic Hsl. The experiments indeed argue in favor of the reviewer’s hypothesis that fat body Hsl is required for optimal fecundity in our setup. The presentation of these data in new Figures 7 panels E and F thus represents a major advance of our study. We did not generate Hsl mutant germline clones since our data are consistent with the conclusion that *Drosophila* mothers transfer Hsl function rather than Hsl-derived lipids to the embryos. This conclusion is illustrated in Figure 5C, which shows that nos-GAL4 driven Hsl expression restores defective SE hydrolase activity to Hsl1 mutant embryos.

2) Mouse data. Previous studies have phenotyped both the adipose tissue of Hsl knockout mice (e.g. PMID: 11717312) and the metabolic phenotypes of mice with an adipose tissue-specific Hsl deletion (PMID: 29232702, which the authors should mention and discuss). In light of these studies, the value of the mouse data presented in this manuscript is rather limited. Given that the authors have access to these mice, it would have made more sense to explore possible effects of disrupted adipose SE catabolism in adipose tissue in vivo, including possible effects on the developing progeny. In this context, the spermatogenesis defects of whole body Hsl knockout mice is intriguing.

We added information about energy and sterol metabolism deficits of different Hsl mutant mouse models to the discussion including the spermatogenesis defects of Hsl mutant males. While the major focus of our manuscript is on *Drosophila* our mouse data, although limited, provide an important proof-of-concept regarding the evolutionary conservation of tissuespecific sterol storage and mobilization. To strengthen this concept, we extended our comparative approach by (1) comparing lipolytic activities of recombinant *Drosophila* Hsl and mouse Hsl in the same experimental setup (new Figure 1A, B), (2) assessing the contribution of murine Hsl to adipose tissue SE hydrolase activities (new Figure 4G), (3) assessing consequences of disturbed adipocyte SE catabolism by measuring gene expression of sterol metabolism genes in adipocyte-specific Hsl knockout mice (new Figure 4H).

We believe that our comparative approach is valuable as it more broadly attracts attention of researchers from different areas and motivate further investigations into the relationship between adipocyte lipolysis and reproduction in both, mammalian and invertebrate models. Assessing a potential intergenerational effect of adipocyte-specific Hsl knockout on mammalian progeny is an exciting future research question but certainly beyond the scope of the presented study.

3) Analyzing endogenous Hsl expression would have been more informative than ectopically expressing it. If the authors are unable provide endogenous expression data, the claims about Hsl localization should be toned down.

We fully agree with the reviewer’s concern. Since we are currently unable to detect endogenous Hsl by immunostaining we changed the wording in the Results section to better reflect the limitations of our over-expression approach.

Minor comments1) In Abstract, the wording "directly equip embryos" is unclear; one needs to read the manuscript to understand what it means.

The Abstract section has been rephrased for clarity.

2) The Introduction extensively described two TG catabolism pathways (bmm and Akhmediated) when, at the end of the day, Hsl turns out not to be the elusive Akh target lipase. Overall, I felt that the authors did a good job at introducing Hsl from a *Drosophila* perspective, but I would have valued a broader introduction about what is (un)known about the Hsl family in other organisms. Similarly, in the discussion, a comparative discussion placing the findings in the context of mammalian literature, cholesterol auxotrophy in flies etc would have been helpful.

We appreciate the notion that the mentioned sections of the introduction and discussion are not well balanced. Accordingly, we included these proposed points in the Introduction and Discussion, respectively. In particular, we added more information about structural similarities of Hsl-related enzymes of different phyla and known physiological functions of mammalian Hsl.

3) In discussion: "and highlight an ancestral function of Hsl-related enzymes in the intergenerational sterol homeostasis of animals." This is a bit misleading because the intergenerational phenotype is only shown in flies.

We agree and have changed the wording accordingly.

4) Some discussion of possible roles for Hsl in males would have been interesting. Are Hsl levels sexually dimorphic/regulated by mating in females?

High-throughput expression data (modENCODE) suggest a moderate mating-dependent Hsl regulation in ovaries but no global sexually dimorphic expression levels. We assessed potentially sexually dimorphic Hsl expression in males, virgin females, and mated females by qRT-PCR but did not find noticeable differences between those groups (data not shown). In the light of the limited value of these results and our limited data on *Drosophila* males we will not further discuss possible sex-specific roles of Hsl.

5) Please provide the dilutions used for the antibodies listed.

Dilutions are now provided in the manuscript.

6) Please clarify how the mutant was validated (genomic PCR?).

Now we included the result of a representative PCR genotyping experiment in the new Figure 2—figure supplement 1 and added the detailed sequence information regarding the molecular characterization of the allele to the Materials and methods section. Moreover, we revised Figure 2A to provide a more precise representation of the Hsl1 allele.

7) In Figure 2G, glycogen levels appear elevated in Hsl mutants but the authors do not discuss this.

We addressed this point in the discussion.

8) In Figure 2E, it would have been nice to also test PAPLA1.

PAPLA1 is part of an ongoing project in our lab and its role in lipid metabolism including a possible interdependency with Hsl will be addressed elsewhere.

9) Should the label of the Y axis in Figure 3D not be 14C-FAs?

The label indicates 14C-SE, which were labeled by 14C-FA precursors. We think that 14C-FA would mislead readers to think “free” 14C-FAs were measured. For clarification we added cartoons to Figure 3D-F indicating the labeled lipid moieties.

10) In Figure 7 and related text it is worth clarifying that the LDM diet already contains some cholesterol prior to cholesterol supplementation.

We changed the wording in the Results section, Materials and methods section, and figure legend to clarify this issue.

11) Please state melanogaster the first time the term *Drosophila* is used.

We now introduce *Drosophila melanogaster* in the Abstract and Introduction.

12) Please include references for the statement regarding lipid mobilisation.

The reference was included.

13) Figure legend 2 states that starvation was monitored every 2-12h, whereas the Materials and methods section states 4-12h. Please clarify.

We apologize for this inconsistency and clarified this issue.

Reviewer #2 (Significance (Required)):

*The study is notable for its depth; it is a careful study of the roles of Hsl in Drosophila which has revealed some surprises (e.g. the SE specificity as well as the intergenerational phenotype). The experiments I proposed would strengthen the latter aspect, which is quite novel, and/or could make it more widely significant*.

[Editors' note: further revisions were suggested prior to acceptance, as described below.]

Specifically, the authors need to emphasize in the discussion that their findings are novel in the fly. In particular, the authors need to temper their statements regarding novelty in the mouse studies. It should also be noted that it is already well appreciated that HSL is a cholesterol ester hydrolase. This is well documented and known for many years (e.g., https://doi.org/10.1186/1743-7075-3-12). The manuscript makes it seem the authors discovered this fact and overstate the advance in this regard. For example, Figure 4E,F is not surprising at all based on this previous knowledge, and they didn't measure DAG which is also greatly increased just like CE is as shown. So the text should be more grounded in what is already known, which is that HSL is already a well know sterol esterase.

We modified several passages of the text including the abstract to clarify that our study was done in flies. We introduced SE hydrolase activity of mammalian Hsl in the Introduction and added information regarding energy and sterol metabolism of different Hsl mutant mouse models to the discussion. We believe that these changes enable a more balanced perception of our study by the scientific community. To avoid overstatements regarding our mouse data we repeatedly state that “Mammalian Hsl is an established regulator of SE breakdown in multiple tissues including testis, adrenal tissue, intestine, and liver.”

In addition, the authors should re-word "Generation of transgenic animals…" etc. to "Generation of transgenic flies…", as they otherwise refer to "fly strains" and the reader might be confused that this is vertebrate (animal) study. Also, statements such as "In summary, our study identifies an ancestral function of Hsl-related enzymes in sterol handling and illustrates how this enzyme orchestrates intergenerational sterol transfer to optimize reproductive success. " should clearly indicate that the work was done in flies.

We changed the statements to indicate that our results refer to flies. We did not re-word “Generation of transgenic animals” since the passage describes the generation of *Drosophila* and mouse models.